# Rising atmospheric moisture escalates the future impact of atmospheric rivers in the Antarctic climate system

Michelle L. Maclennan [1,2] ✉, Andrew C. Winters [1], Christine A. Shields [3], Rudradutt Thaker [4], Léonard Barthelemy [5], Francis Codron [5] & Jonathan D. Wille [6]

Despite their relative rarity, atmospheric rivers are key contributors to the surface mass balance of Antarctica. However, the future role of atmospheric rivers in modulating Antarctic sea-level contributions is a major area of uncertainty. Here, we leverage high-resolution climate simulations to show that Antarctic atmospheric rivers are highly sensitive to future increases in atmospheric moisture, leading to a doubling of atmospheric river frequencies and 2.5 × increase in precipitation from 2066–2100 under present-day thresholds for atmospheric river detection. However, future precipitation impacts are critically dependent on the detection threshold: accounting for moisture increases in the threshold produces smaller, regional changes in atmospheric river frequency, primarily resulting from an eastward shift in the polar jet maximum wind speeds. Our results underscore the importance of using large ensembles to quantify Antarctic atmospheric river responses to variability in projected moisture, which may not be captured when using only a few ensemble members.

The Antarctic Ice Sheet plays a key role in our climate system by modulating present and future global mean sea level through the accumulation, storage, and discharge of ice over time. In the last four decades, mass loss from Antarctica has increased by a factor of six, driven by ice shelf basal melting and resultant grounding line retreat in West Antarctica, leading to an acceleration of ice discharge into the ocean[1]. Future simulations for Antarctic mass balance project mass losses of up to 150 cm sea level equivalent per year by 2100[2].

Precipitation is crucial in offsetting the vast majority of mass loss from Antarctica to the ocean each year [91%,[1,3]]. In the future, net precipitation in medium-high emissions scenarios is projected to increase the Antarctic surface mass balance by 20–30%[4–6]. In many regions of the continent, up to 40% of net precipitation and 70% of precipitation variability is driven by synoptic-scale extreme events[7–9]. However, there are few studies that examine how these precipitation extremes may be impacted by anthropogenic warming, despite indications that synoptic events will remain the main driver of snow accumulation and that rainfall (which mostly occurs during extreme weather events) may increase by a factor of two to five by the end of the twenty-first century[10,11]. The Antarctic Ice Sheet is the greatest source of uncertainty for future mean sea level projections[2], and while

precipitation increases may not compensate for future accelerations in ice discharge, quantifying future changes in extreme precipitation events is essential to constraining estimates for global mean sea level rise in the twenty-first century and beyond.

Here, we focus on the most intense mechanisms for poleward moisture transport in the atmosphere, known as atmospheric rivers [ARs;[12]]. Despite occurring only 1% of the time at any given location along the Antarctic coastline, these narrow, elongated filaments of extreme water vapor transport are associated with 13% of the annual total precipitation and 35% of interannual variability in precipitation over Antarctica[9,13]. Although snowfall is the dominant impact of ARs on the Antarctic Ice Sheet in the present-day, ARs are also associated with the majority of extreme surface melt events in West Antarctica and on the Antarctic Peninsula, contributing to the destabilization of buttressing ice shelves[14]. Recent studies based on global AR detection tools suggest the thermodynamics (increases in atmospheric moisture capacity through the Clausius-Clapeyron effect) drive strong AR frequency increases in future climates[15–19]. However, global AR detection tools struggle to capture Antarctic ARs[20], and as a result, little is known about the thermodynamic and dynamical responses of Antarctic ARs to anthropogenic

[1]Department of Atmospheric and Oceanic Sciences, University of Colorado Boulder, Boulder, CO, USA. [2]British Antarctic Survey, Cambridge, UK. [3]Climate and Global Dynamics Lab, NSF National Center for Atmospheric Research, Boulder, CO, USA. [4]Department of Atmospheric and Oceanic Sciences, University of Wisconsin-Madison, Madison, WI, USA. [5]Laboratoire d'Océanographie et du Climat (LOCEAN/IPSL), Sorbonne Université; CNRS; MNHN; IRD, Paris, France. [6]Institute for Atmospheric and Climate Science, ETH Zurich, Zürich, Switzerland. ✉e-mail: michelle.maclennan@colorado.edu

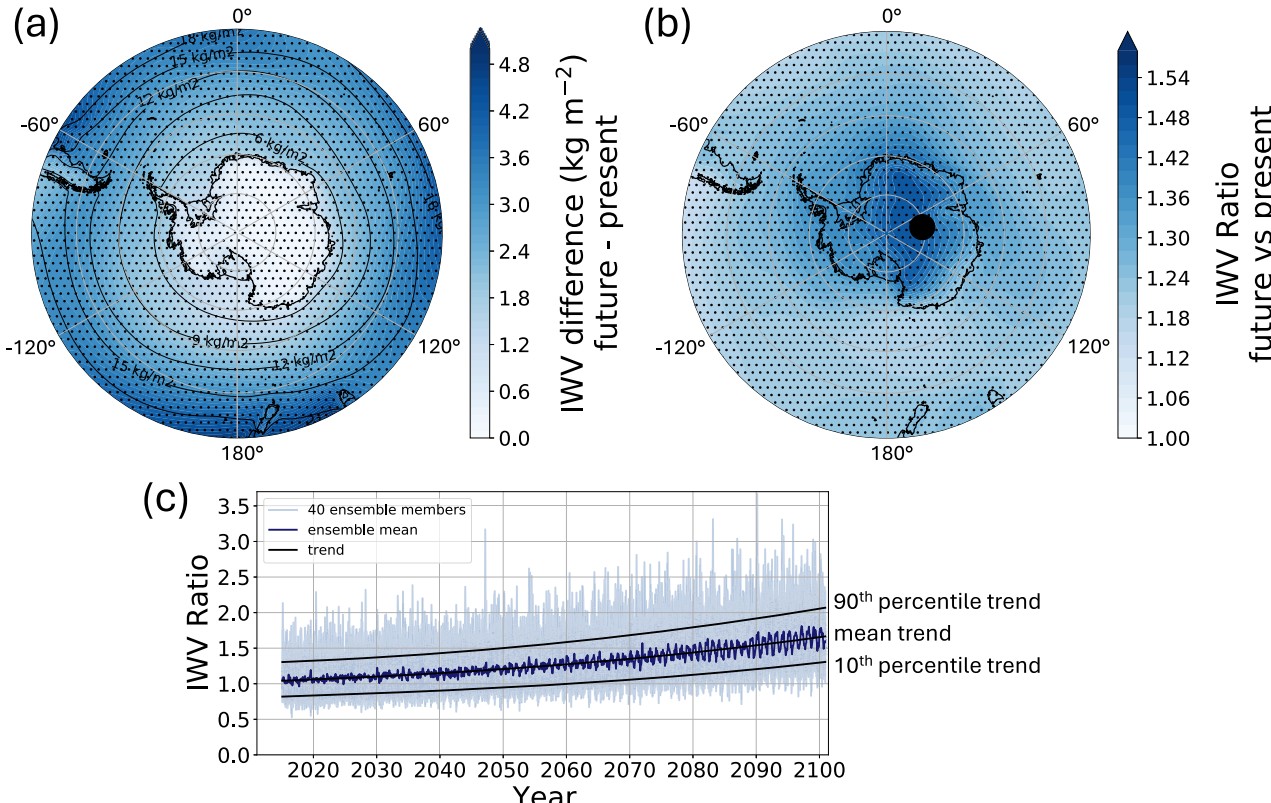

**Fig. 1 | Twenty-first century increase in atmospheric moisture. a** Mean absolute increase and **b** mean relative increase in integrated water vapor (IWV) at each grid point from 1980–2014 to 2066–2100 in the CESM2. Stippling indicates regions where the mean change in IWV among ensemble means exceeds the standard deviation. **c** Time series showing the increase in IWV between 2015 and 2100 relative to the 1980–2014 monthly climatology over the East Antarctic Plateau at 80.6° S, 80°

E (marked as black dot on map in (**b**), and representative of IWV increases across the interior of the Antarctic Ice Sheet). In light blue, the relative increase in IWV of all 40 CESM2 members; in dark blue, the ensemble mean; in black, the 3rd-degree polynomial trend lines derived from a 10-year running mean of the ensemble mean, 10th percentile, and 90th percentile curves—representing the curves used to scale the threshold for future AR detection.

warming, despite the critical importance of AR-attributed precipitation for the Antarctic climate system.

In this study, we leverage 40 high-resolution ensemble members from the Community Earth System Model, version 2 (CESM2) large ensemble[21,22] to assess how Antarctic AR frequencies and precipitation impacts change in the twenty-first century under the Shared Socioeconomic Pathway 3-7.0 radiative forcing scenario (SSP3-7.0). We apply a polar-specific AR detection tool based on the poleward component of integrated water vapor transport (vIVT) to CESM2 over 1980–2014 and 2066–2100[13]. To dissect the thermodynamic and dynamical drivers for future AR frequencies, we scale the present-day AR detection threshold by the relative increase in integrated water vapor (IWV) at each point over the Southern Ocean and Antarctica, which is shown to be a robust method to account for the Clausius-Clapeyron effect in AR detection[15]. We then quantify the changes to AR-attributed total precipitation and rainfall over Antarctica, demonstrating the extreme sensitivity of AR precipitation impacts to the detection threshold. Ultimately, this study presents a crucial analysis of the impacts of anthropogenic warming on Antarctic ARs, and how these impacts, in turn, affect AR-attributed precipitation over the Antarctic Ice Sheet and the way we describe the role of ARs in the Antarctic climate system.

## Results

### Atmospheric moisture increase

An exponential increase in atmospheric moisture occurs over the Southern Ocean and Antarctica over the twenty-first century relative to the present-day, strengthening the pole–equator moisture gradient (Fig. 1a). The greatest increase in ensemble mean integrated water vapor

(IWV, +5 kg m$^{-2}$) occurs over the Atlantic Ocean east of South America, the Indian Ocean equatorward of the Amery Ice Shelf, and the Pacific Ocean equatorward of the Ross Sea (see Fig. S1 for map of locations). The ensemble mean IWV increases less than 1 kg m$^{-2}$ over the Antarctic continent (Fig. S2e–h), which is characterized by extremely cold and dry air. However, this region is marked by the greatest relative increase in IWV (1.5 × present-day) at the end of the twenty-first century (Fig. 1b).

There is large variability (2 kg m$^{-2}$) in the future IWV increase among the 40 ensemble members, with extremely high IWV ratios during individual months from 2015 to 2100, which can reach 3–3.5 times the present-day climatology by the end of the twenty-first century (Fig. 1c). The 10th and 90th percentiles of IWV increase range from 1.3–2.1 × present-day values, demonstrating uncertainty in the magnitude of the relative future increase in atmospheric moisture even within the single radiative forcing scenario of SSP3-7.0, and the advantage of using many ensemble members to capture this range of variability.

### Atmospheric river frequencies in the present and future climates

AR frequencies in CESM2 are comparable to MERRA-2 reanalysis in the present-day, with maximum frequencies of 4 days yr$^{-1}$ over the Southern Ocean, 3.5 days yr$^{-1}$ in coastal West Antarctica, Dronning Maud Land, and Enderby Land, and less than 1 day yr$^{-1}$ over the East Antarctic Plateau (Fig. 2a). CESM2 exhibits a positive bias in AR frequency of 0.5 days yr$^{-1}$ offshore of West Antarctica and over the Ross Ice Shelf (Figs. 2b and S3), which may be explained by a low-pressure bias of CESM2 in this region, conducive to poleward moisture transport (Fig. S4a–d). CESM2 exhibits a negative bias in AR frequency of −0.5 days yr$^{-1}$ over western Dronning Maud Land and the East

**Article**

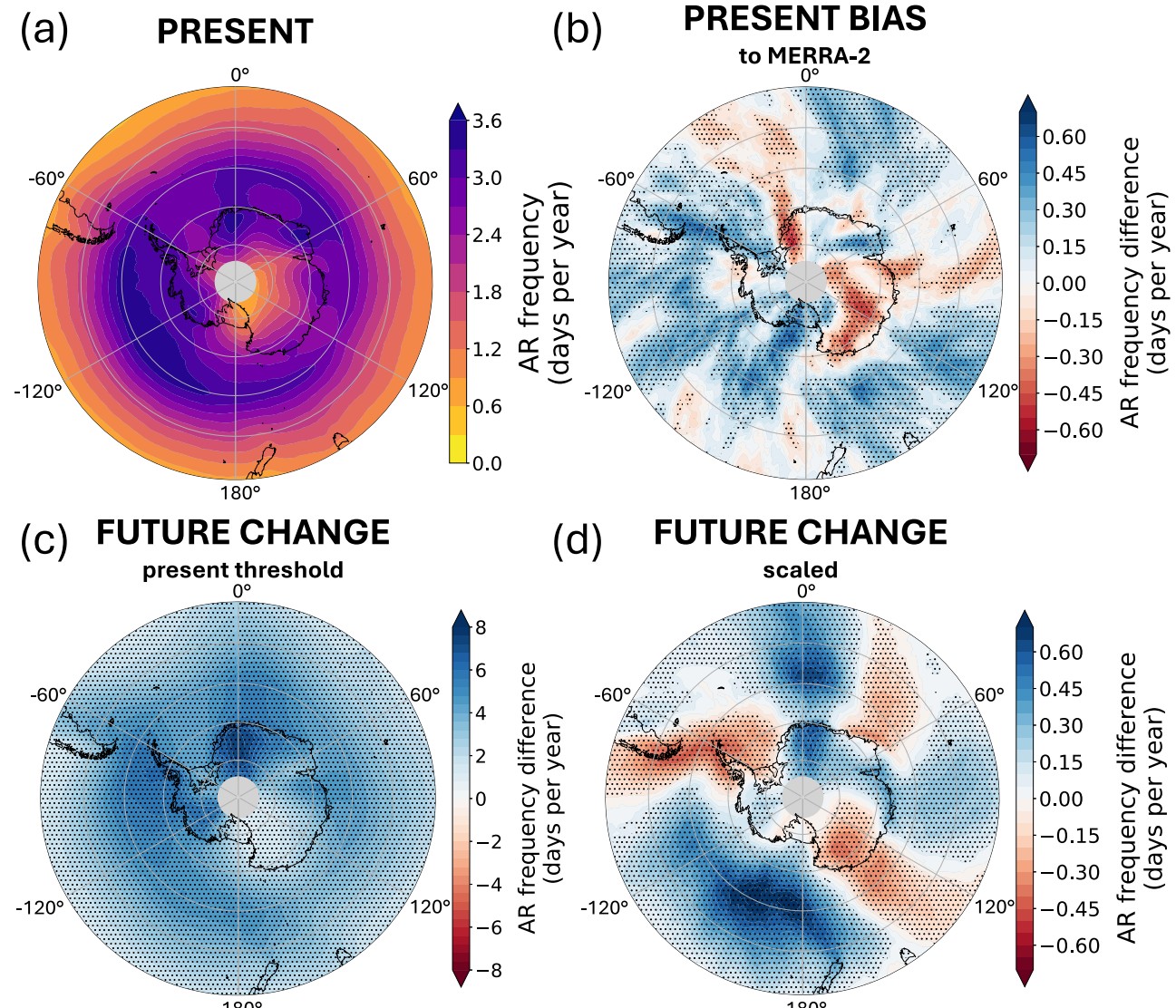

**Fig. 2 | Present and future frequency of atmospheric rivers. a** Present-day average annual AR frequency among all 40 ensemble members in CESM2 (1980–2014). **b** Model AR frequency bias (CESM2 minus MERRA-2 reanalysis) in the present (1980–2014). **c** Future AR frequency difference in CESM2 (2066–2100 minus 1980–2014) when future AR detection is solely based on the present-day vIVT threshold. **d** Future AR frequency difference in CESM2 (2066–2100 minus 1980–2014) when the present-day threshold for AR detection is scaled by the ensemble mean relative increase in IWV. In **b**, **c**, **d**, stippling indicates regions where the mean difference in AR frequency exceeds the standard deviation among the 40 CESM2 ensemble members.

Antarctic Plateau, directly poleward of regions with negative mean sea level pressure biases—suggesting that stronger or more frequent cyclones in CESM2 may act to block poleward flow towards the ice sheet, producing lower AR frequencies relative to MERRA-2 in these regions.

When applying the present-day threshold for AR detection (based on the 98th percentile of vIVT) to the future period in CESM2, AR frequencies double by the end of the century, with the ice sheet-integrated AR frequency increasing from 15.0 ± 0.3% (1980–2014) to 31.7 ± 0.5% (2066–2100). AR frequencies increase up to 6 days yr$^{-1}$ over the Southern Ocean and West Antarctica and 8 days yr$^{-1}$ over Dronning Maud Land (Fig. 2c). These large, widespread increases in AR frequency suggest a substantial response to future changes in climate compared to the present day. To test whether this response is thermodynamically or dynamically driven, we raise the AR detection threshold by the relative increase in IWV (ensemble mean and 10th and 90th percentiles) at the end of the twenty-first century to account for moisture increases (Fig. S5a–c). With ensemble mean IWV scaling, AR frequencies from 2066–2100 are comparable to the present-day (Fig. 2d). However, there are three distinct regions of alternating AR frequency

increase and decrease, with increases offshore West Antarctica (0.6 days yr$^{-1}$), over Dronning Maud Land (0.2–0.5 days yr$^{-1}$), and the Amery Ice Shelf (0.2 days yr$^{-1}$), and decreases over the Antarctic Peninsula (−0.4 days yr$^{-1}$), offshore Enderby Land (−0.1 day yr$^{-1}$), and Wilkes Land (−0.2 days yr$^{-1}$). It is clear from the difference between Fig. 2c and d that moisture changes dominate increases in Antarctic AR frequency at the end of the twenty-first century compared to the present day, while changes in atmospheric circulation play a secondary role in producing a small, zonal wave three-like change in AR frequencies[23].

## Impacts on Antarctic precipitation

In the present-day (1980–2014), ARs contribute 418 ± 29 Gigatons (Gt) yr$^{-1}$ of precipitation (1.15 mm sea level equivalent, or 13 ± 0.6 % of the annual total precipitation) over the Antarctic Ice Sheet in CESM2. AR precipitation is highest in coastal Antarctica (100 mm w.e. yr$^{-1}$) and decreases rapidly from the coast towards the interior (10 mm w.e. yr$^{-1}$), which is seldom reached by ARs (Fig. 3a, b). ARs explain 30–50% of the interannual variability in the total precipitation from 1980–2014 over most of the ice sheet, except for the East Antarctic Plateau and the Antarctic Peninsula (Fig. 3c).

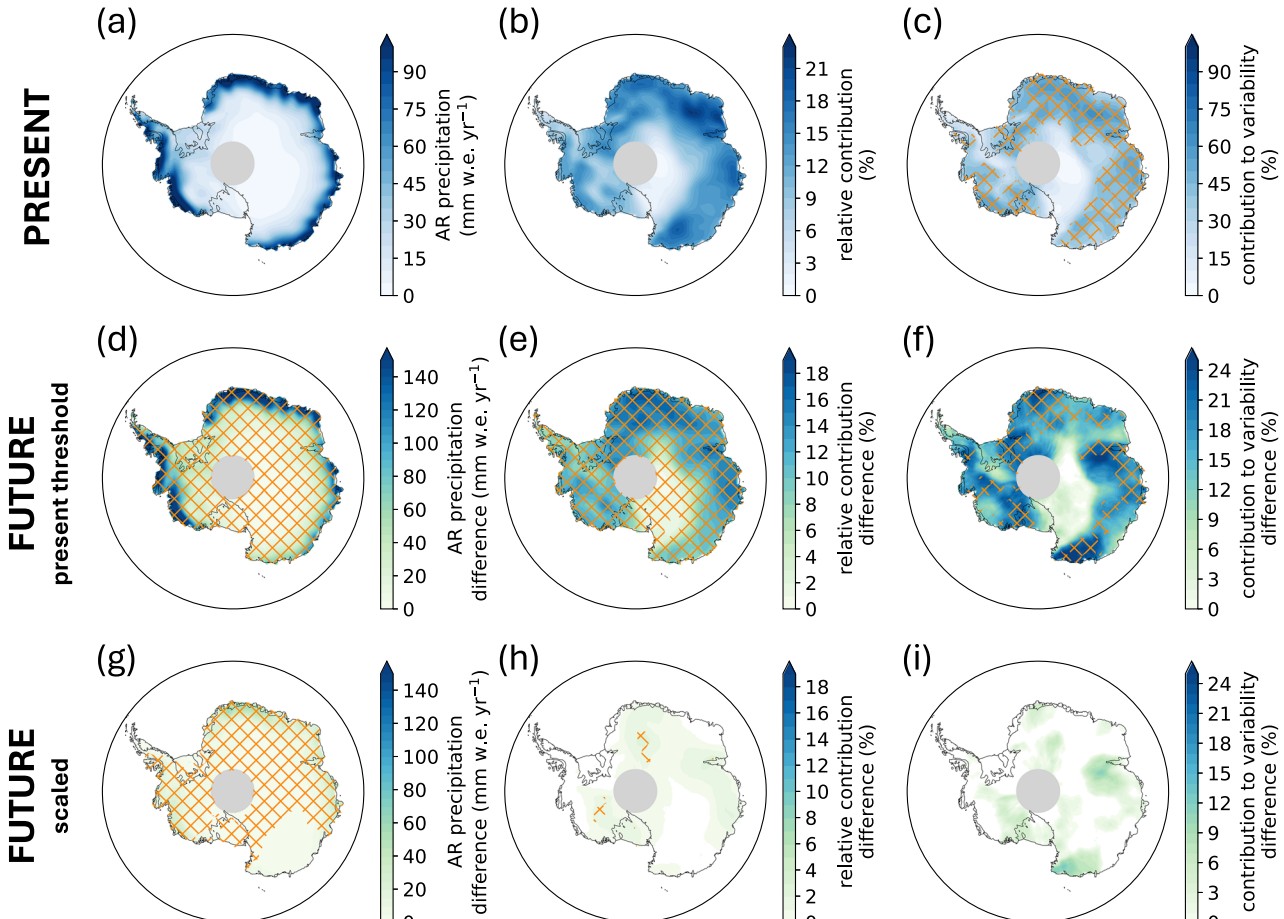

**Fig. 3 | Future changes in atmospheric river-attributed total precipitation.**
CESM2 annual average AR-attributed precipitation (snowfall and rainfall) in
**a–c**, the present, **d–f** the future change in AR-attributed precipitation with
AR detection based on the present-day threshold, and **g–i** the future change in
AR-attributed precipitation when the AR detection threshold is scaled for the
atmospheric moisture increase. **a** Annual average AR-attributed precipitation over
1980–2014. (**b**) the relative contribution of AR precipitation to the total annual
precipitation over 1980–2014. **c** the relative contribution of ARs to the interannual
variability in total precipitation, with hatching indicating regions of statistically

significant contributions to variability ($p < 0.1$). **d, g** Absolute change in AR pre-
cipitation, **e, h** the change in the relative contribution of AR precipitation to total
precipitation, and (f and i) the change in contribution of AR precipitation to year-to-
year variability in total precipitation between the future (2066–2100) and the present
(1980–2014). For **d, e, g, h**, hatching indicates regions of statistically significant
change (where the mean change among the 40 ensemble members exceeds their
standard deviation). For **f, i**, hatching is applied to regions of statistically significant
change where AR-attributed precipitation also exhibits a statistically significant
contribution to variability ($p < 0.1$).

---

CESM2 exhibits a positive bias in total precipitation (up to 450 mm w.e.
yr$^{-1}$) and AR precipitation (up to 80 mm w.e. yr$^{-1}$) relative to MERRA-2,
particularly in coastal Dronning Maud Land and Enderby Land (Fig. S6).
However, CESM2 puts less importance on AR-related precipitation than
MERRA-2: the bias in the relative contribution of ARs to the total pre-
cipitation ranges from $-7\%$ (Wilkes Land) to 9% (interior Dronning
Maud Land).

Using the present-day vIVT threshold for future AR detection, we find
that AR precipitation increases 2.5x over Antarctica over 2066–2100 to
1012 $\pm$ 138 Gt yr$^{-1}$ (2.8 mm sea level equivalent), with all coastal regions
except Wilkes Land experiencing increases up to 150 mm w.e. yr$^{-1}$ (Fig. 3d).
Increases in AR precipitation outpace increases in total precipitation
(Fig. S7), such that the relative contribution of ARs to the total rises to 24 $\pm$ 2
% (ice sheet-integrated value, see Fig. 3e for regional changes), and the
contribution of ARs to precipitation variability increases by 25% in Dron-
ning Maud Land, the Amery Ice Shelf region, Wilkes Land, and parts of
West Antarctica (Fig. 3f). Comparatively, when scaling for the mean relative
increase in IWV in the detection of future ARs, AR precipitation increases
by only 1.25× (536 $\pm$ 27 Gt yr$^{-1}$ or 1.5 mm sea level equivalent, Fig. 3g). The
relative contribution of ARs to the total precipitation remains at 13 $\pm$ 0.3 %,
and there is no significant change in the contribution of ARs to precipitation
variability (Fig. 3h, i). Scaling by the 10th and 90th percentile of IWV

increase yields an increase of 100–200 mm w.e. yr$^{-1}$ in AR precipitation and
an increase in their relative contribution of 10–30% by region (Fig. S8).
Thus, when accounting for the increase in atmospheric moisture in the
detection of Antarctic ARs, the relative contribution of ARs to total pre-
cipitation is comparable to the present-day, with modest increases in future
AR precipitation keeping pace with increases in the total precipitation. This
creates a large discrepancy with projected precipitation impacts when only
the present-day threshold for AR detection is applied to the twenty-first
century, which results in a tremendous increase in AR precipitation across
the ice sheet, and particularly in coastal regions and on ice shelves. This
disparity, driven by the sensitivity of AR detection to increases in atmo-
spheric moisture, has an enormous impact on how we describe the relative
importance of ARs in the Antarctic climate system and their future con-
tributions to the surface mass balance of the ice sheet.

In the present-day, ARs contribute up to 25 mm w.e. yr$^{-1}$ of rainfall in
coastal Antarctica (up to 30% of the total rainfall, Fig. 4a, b), particularly
along the Bellingshausen Sea, and explain 40–75% of rainfall variability
across most major ice shelves (Fig. 4c). However, CESM2 exhibits a large
positive bias in the total rainfall (up to 175 mm w.e. yr$^{-1}$) and AR rainfall (up
to 25 mm w.e. yr$^{-1}$) on the Bellingshausen and Antarctic Peninsula ice
shelves (Fig. S6). These present-day biases may inflate future projections of
liquid precipitation over the Antarctic Peninsula in CESM2 under SSP3-7.0,

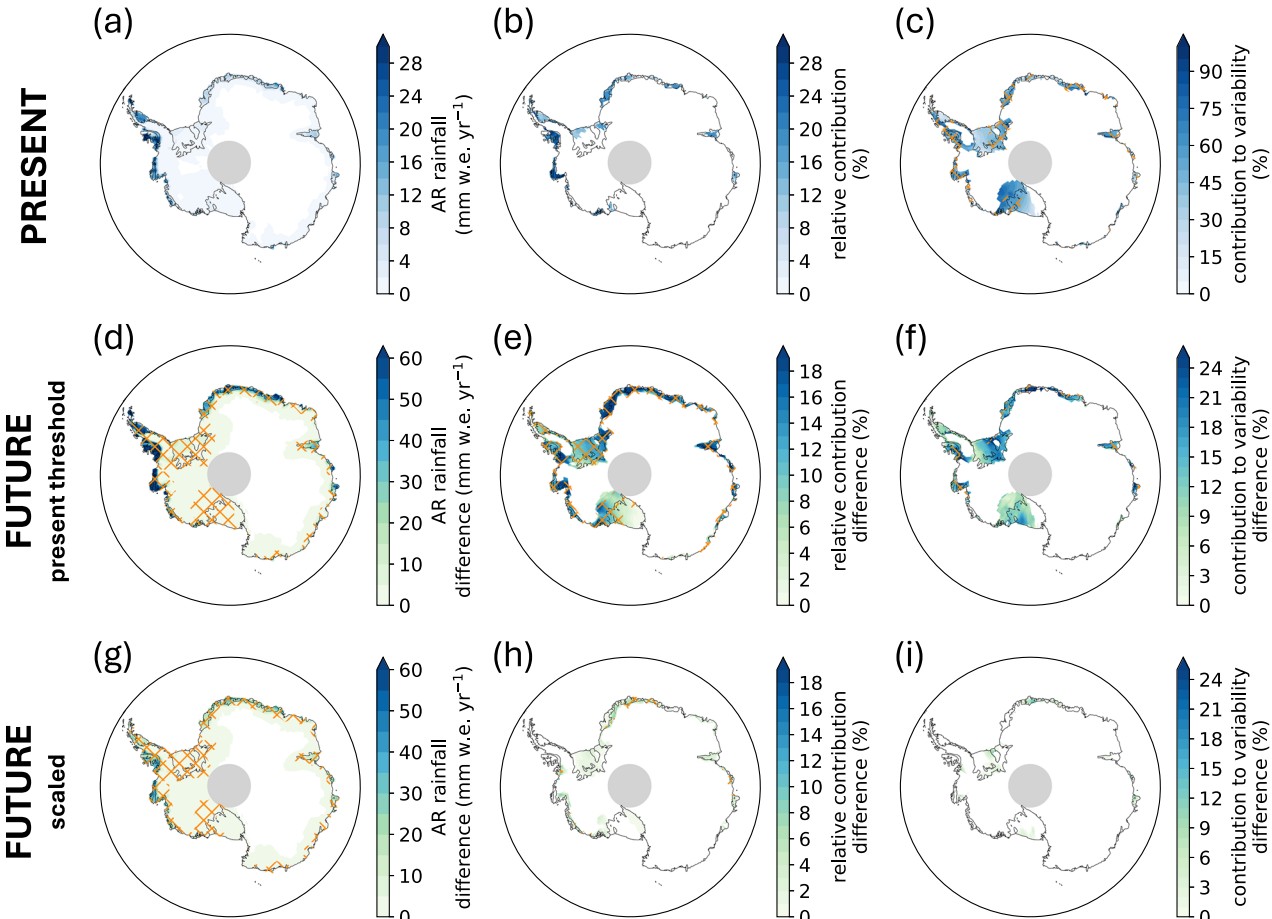

**Fig. 4 | Future changes in atmospheric river-attributed rainfall.** CESM2 annual average AR-attributed rainfall in **a–c** the present, **d–f** the future change in AR-attributed rainfall with AR detection based on the present-day threshold, and **g–i** the future change in AR-attributed rainfall when the AR detection threshold is scaled for the atmospheric moisture increase. (a) annual average AR-attributed rainfall over 1980–2014. **b** the relative contribution of AR rainfall to the total annual rainfall over 1980–2014. **c** the relative contribution of ARs to the interannual variability in total rainfall, with hatching indicating regions of statistically significant contributions to variability ($p < 0.1$). **d, g** Absolute change in AR rainfall, **e, h** the change in the relative contribution of rainfall to total rainfall, and (f and i) the change in contribution of AR rainfall to year-to-year variability in total rainfall between the future (2066–2100) and the present (1980–2014). For **d, e, g, h**, hatching indicates regions of statistically significant change (where the mean change among the 40 ensemble members exceeds their standard deviation). For **f, i**, hatching is applied to regions of statistically significant change where AR-attributed rainfall also exhibits a statistically significant contribution to variability ($p < 0.1$).

and therefore estimates of future ice shelf surface mass balance and firn air content reliant only on output from the CESM2 large ensemble may overestimate damage to the snowpack by rainfall. The relative contribution of ARs to the total rainfall in CESM2 is lower than MERRA-2 by more than 10% over Antarctic Peninsula ice shelves, indicating the role of ARs in contributing rainfall to this region is reduced in CESM2 compared to MERRA-2.

With the present-day vIVT threshold for future AR detection, AR rainfall increases exceed 60 mm w.e. yr$^{-1}$ over the Bellinghausen coast, Larsen C Ice Shelf, and coastal Dronning Maud Land (Fig. 4d). The relative increase in AR rainfall outpaces increases in total rainfall (Fig. S7), such that the contribution of ARs to total rainfall rises from 21–34% (Fig. 4e), and the contribution to rainfall variability increases by 15–20% over the Bellingshausen coast, coastal Dronning Maud Land, and the Amery region (Fig. 4f). When scaling the threshold for the mean IWV increase, increases in AR rainfall are more modest (up to 40 mm w.e. yr$^{-1}$ over the Bellingshausen coast and coastal Dronning Maud Land) and there is little change in the relative contribution of ARs to total rainfall, nor in their relative contribution to rainfall variability (Fig. 4g–i). Changes in AR rainfall and the contribution of ARs to total rainfall are sensitive to the method of scaling, particularly in coastal areas, where scaling by the 10th and 90th percentile of the IWV increase yields a range of AR rainfall increase of 30–50 mm w.e. yr$^{-1}$ (15–30% relative contribution) by the end of the twenty-first century

(Fig. S9). Despite rainfall being a small component of the total precipitation in Antarctica, in both the present and future climates up to the year 2100, these results highlight the increasingly important role of extreme, synoptic systems in producing rainfall over coastal Antarctica and its ice shelves under the SSP3-7.0 warming scenario. Given the compounding impacts of rainfall in Antarctica, including the raising of firn temperatures through refreezing of liquid water, reduction of surface albedo, firn air depletion, and ice erosion – all of which precondition the surface for widespread melting[24]—the increase in AR-associated rainfall may have a considerable impact on future ice shelf stability.

## Linking circulation changes to atmospheric river activity

When the detection threshold is scaled by the relative increase in IWV at each grid point, future AR frequencies are comparable to the present day, but with a small, zonal wave three-like pattern of change (Fig. 2d). Here, we break down these changes in AR frequency by season and compare them with major changes in the large-scale atmospheric circulation from 2066–2100 relative to 1980–2014.

In austral summer (Fig. 5a; see Figs. S10–S15 for bias and change in individual variables), increases in the 500 hPa geopotential height are mirrored by increases in mean sea level pressure over South America and offshore Dronning Maud Land and Wilkes Land. We find increased 850 hPa poleward winds to the west of each region, co-located with higher AR

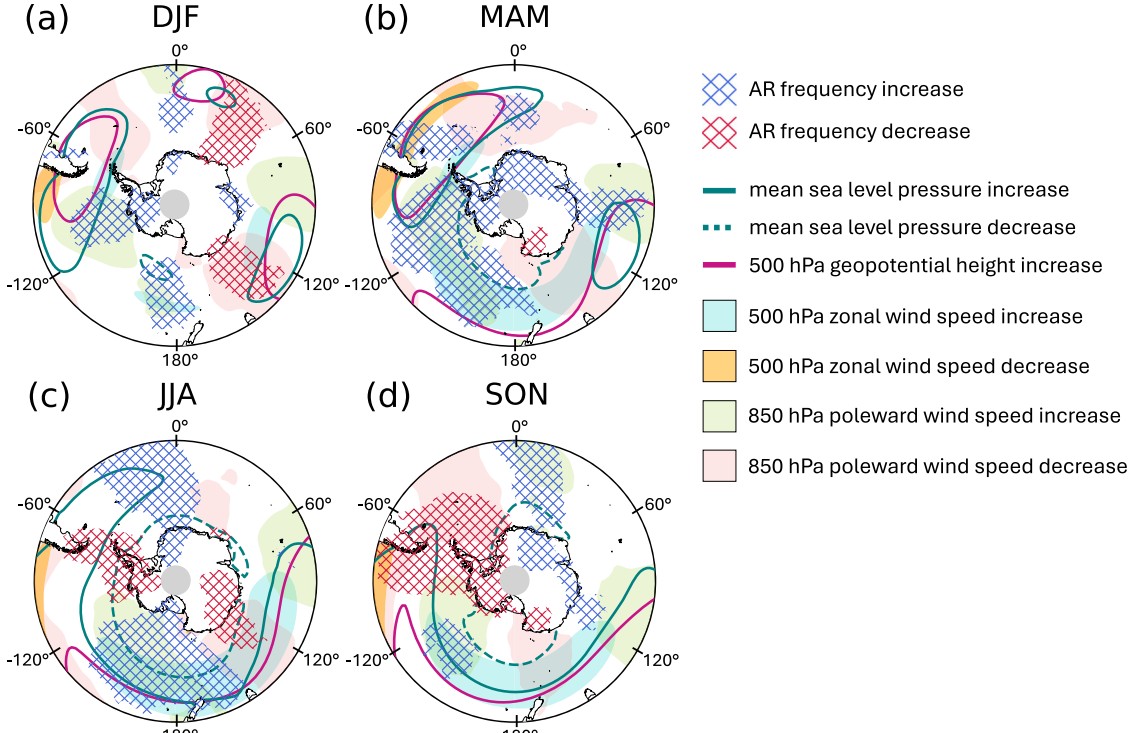

**Fig. 5 | Future changes in atmospheric circulation and atmospheric river frequency by season.** CESM2 ensemble mean difference in AR frequency and atmospheric circulation from 1980–2014 to 2066–2100 (future minus present) in **a** austral summer (December–January–February), **b** austral fall (March–April–May), **c** austral winter (June–July–August), and **d** austral spring (September–October–November). Red and blue cross-hatching indicates a change in AR frequency of 0.4 days per year or greater, pink contours indicate a change in the 500 hPa geopotential height of 90 m or greater (there are no regions of decrease in geopotential heights), and teal contours indicate an increase (solid line) and decrease (dashed line) in mean sea level pressure exceeding 1.8 hPa. Light blue and orange shading indicate a change in the 500 hPa zonal wind speed of 2 ms$^{-1}$ or greater (light blue for increase, orange for decrease) while green and pink shading indicate a change in the poleward 850 hPa meridional wind speed of 0.3 ms$^{-1}$ or greater (green for increase, pink for decrease). For each variable, only regions of statistically significant change are plotted, based on a two-tailed z-statistic at the 95% confidence interval.

frequencies in the South American and Dronning Maud Land regions, and decreased 850 hPa poleward winds to the east of each region, co-located with decreased AR frequencies in Dronning Maud Land and Wilkes Land. Sea ice extent decreases are largest near the Antarctic Peninsula and Dronning Maud Land (Fig. S15i), but are not closely aligned with AR frequency changes.

In austral fall (Fig. 5b), there is a widespread increase in AR frequency over the Southern Ocean and Antarctica, which may be related to large increases in IWV (though high-moisture biases in CESM2 may lead to an overestimation of the future change in IWV in this season – Fig. S2). 500 hPa geopotential heights increase over the Southern Ocean from 90°E to 110°W (mirrored by higher mean sea level pressure), strengthening the pole-to-midlatitude geopotential height gradient and inducing an eastward shift of the maximum wind speeds in the polar jet stream. As a result, we find deeper mean sea level pressures over the Ross Sea and enhanced poleward flow and increased AR frequencies to the east of the surface low, co-located with the shifted jet exit region. The largest decreases in sea ice are concentrated near the Antarctic Peninsula (Fig. S15j), co-located with an AR frequency increase.

In austral winter (Fig. 5c), we observe an even greater strengthening of the height gradient and associated eastward shift of the polar jet first observed in fall, with increased poleward flow and AR frequencies in the jet exit region at 170° W, and decreased poleward flow and AR frequencies at 130° E (directly poleward of the jet). AR frequencies decrease over the Antarctic Peninsula, likely resulting from a poleward shift of the storm track and enhanced zonal flow over the Peninsula (Fig. S13g), exacerbating present-day biases in the AR detection tool, which is tuned to capture meridionally-propagating ARs. Therefore, we note that AR detection

methods based on IWV or total IVT (both meridional and zonal water vapor transport) may produce better classifications in this region when considering the effect of circulation changes on future AR frequency[20]. Offshore of Dronning Maud Land, there is an increase in 850 hPa poleward winds and AR frequencies co-located with a major sea ice decline and increased 2 m temperatures (Figs. S15k, S12g, and S16).

In austral spring (Fig. 5d), similar patterns in the 500 hPa geopotential height and polar jet persist from winter, as well as the increase in AR frequencies and major sea ice decline offshore of Dronning Maud Land (Fig. S15l).

## Discussion

Global studies of ARs in future climates identify the greatest source of uncertainty in future AR frequency to be the choice of AR detection method, and in particular, whether the detection threshold accounts for the Clausius-Clapeyron effect, which dominates future changes in AR frequency[15–18]. While we apply only one AR detection algorithm in this study, shown to be the most effective at capturing ARs that penetrate the interior of ice sheet[20], we test the sensitivity of future Antarctic AR frequencies and precipitation to increasing atmospheric moisture through four different thresholds (the present-day 98th percentile of vIVT and three scaled thresholds based on the ensemble mean IWV increase and the 10th and 90th percentiles). Under the present-day threshold, we find a doubling of AR frequency and a 2.5 x increase in precipitation by the end of the twenty-first century. Comparatively, when scaling for the mean increase in IWV, AR frequencies are comparable to the present-day, with a small increase in precipitation, though an eastward shift of the polar jet and southward shift of the storm track produce a zonal wave three-like regional shift in AR activity.

Our results show that the future role of ARs in the Antarctic climate system is extremely vulnerable to the detection method chosen, which has serious implications for quantifying the influence of extreme events on Antarctic contributions to global mean sea level rise. Raising the threshold for AR detection may better preserve the structural consistency (filamentous shape) of ARs in future climates[15]. However, using the scaled threshold for AR detection to conclude that future ARs retain a comparable role in the Antarctic climate system compared to the present-day undermines the dramatic future increases in AR snowfall and rainfall observed with the present-day threshold (caused by events that would be considered ARs today, but do not meet the future scaled threshold). This suggests the choice of future AR threshold is a science-specific decision that depends on the systems considered, just like the choice of AR detection tool[20]. Our findings are largely consistent with global studies of future ARs, which suggest a strong response of ARs to thermodynamic forcing in high emission scenarios resulting in a poleward shift in AR frequency in the Southern Hemisphere[19]. Furthermore, global studies confirm ocean basin-specific non-linearities in moisture increase[18], highlighting the need to use large ensembles of global climate models and a grid cell-specific approach for moisture scaling, as this scale of variability may not be captured when using only a few ensemble members. Finally, while 1˚ model grid spacing is widely used for AR detection in future climates and is capable of accurately capturing large-scale features[16,25], a finer spatial resolution would provide higher-fidelity representations of AR structure, interactions with topography, and surface impacts[26], motivating the application of regional climate modeling for future AR studies.

While the CESM2 large ensemble provides a range of uncertainty in future AR projections, we acknowledge that CESM2 model biases may enhance localized patterns of future climate change. The present-day overestimation of Antarctic precipitation by CESM2 may exaggerate future increases in AR-driven snowfall and rainfall[4,27]. The positive bias in precipitation is driven at least in part by the underestimation of sea ice area in CESM2 in the present-day, which promotes enhanced evaporation from the ocean surface and a resulting overestimation of snowfall over the ice sheet[28]. Nevertheless, multi-model analyses suggest robust increases in Antarctic coastal precipitation (up to a 30% increase in total precipitation, and a doubling or tripling of rainfall) at the end of the twenty-first century in high emissions scenarios, consistent with the findings presented here[11,27]. CESM2 also exhibits a pronounced low 500 hPa height bias around Antarctica, producing an eastward bias in the maximum zonal wind speeds of the polar jet in the present-day, which may amplify its intensification in future climates. However, the future increases in the 500 hPa geopotential height gradient, eastward-shifted jet stream, and resultant surface low pressure system intensification in West Antarctica (the Amundsen Sea Low) are remarkably consistent with CMIP5 projections among multiple global climate models[29]. Enhanced poleward flow in the jet exit region is co-located with an increase in AR frequency, increased precipitation at the coast, and a reduction in sea ice offshore – all of which are linked to ocean warming in the Amundsen Sea and resultant grounding line retreat of West Antarctic glaciers[30]. In East Antarctica, the co-location of dynamically-driven increases in AR frequency with strong sea ice decline offshore Dronning Maud Land in austral winter and spring may signal a positive feedback loop where reduced sea ice extent enhances moisture availability for Antarctic ARs, while the increased frequency of ARs and associated downward longwave radiation and rainfall exacerbates sea ice melting[31]. This suggests that future increases in AR activity may initiate complex atmosphere-ice-ocean feedbacks in Antarctica. Ultimately, how we detect Antarctic ARs and attribute precipitation in the present-day, and how we adapt these methodologies to future climate states, will determine how we describe the importance of ARs in the Antarctic climate system.

## Methods
### Datasets
The CESM2 large ensemble consists of 100 ensemble members spanning 1850–2100[21,22]. CESM2 is a fully coupled atmosphere-ocean-land model

and uses the Community Atmosphere Model Version 6 (CAM6) with 32 vertical levels and a model top at 2.26 hPa (about 40 km). Here we use 40 ensemble members from the CESM2 large ensemble with a high (6-hourly) temporal resolution and 1.25° longitude by 0.95° latitude horizontal grid spacing. We compare the present-day period from 1980 to 2014 (with CMIP6 historical forcing) to future projections from 2015 to 2100 under Shared Socioeconomic Pathway 3-7.0 (SSP3-7.0) forcing. The SSP3-7.0 scenario is broadly used in CMIP6 model intercomparisons and represents an important pathway of mid-high range warming by 2100 that builds on RCP8.5[32,33]. Standard CMIP6 simulations use biomass burning emissions based on remote sensing data from 1997 to 2014, which introduces greater interannual variability in simulations and can impact future projections of the climate response to atmospheric forcing[34,35]. We use ensemble members with smoothed biomass burning (SMBB), which uses a biomass burning protocol from 1997–2014 that is more comparable to emissions used prior to 1997. From CESM2, we use vIVT to detect ARs, IWV, 2 m temperature, and sea ice fraction to examine anthropogenic warming-driven changes in climate, and total precipitation and surface skin temperature to partition precipitation into snowfall and rainfall while quantifying AR impacts on Antarctica. Furthermore, we use 500 hPa geopotential height, mean sea level pressure, 500 hPa zonal winds and 850 hPa meridional winds to quantify changes in the atmospheric circulation around Antarctica.

To understand model uncertainty, we briefly compare the CESM2 large ensemble to the U.S. Department of Energy's Energy Exascale Earth System Model (E3SMv2) large ensemble, another CMIP-class Earth System Model with a similar and atmosphere component but completely different ocean, both in physics and numerical grid[36]. E3SMv2 large ensemble is a 21-member ensemble and employs CMIP6 historical forcing using a similar initialization approach to portions of the 100 member CESM2 large ensemble. Further details on the ensemble design, model, and the comparison between CESM2 and E3SMv2 can be found[37]. Because the E3SMv2 large ensemble is smaller than the CESM2 large ensemble, we subset CESM2 to the same ensemble size as E3SMv2 for consistency.

We employ the Modern-Era Retrospective analysis for Research and Applications, version 2 (MERRA-2)[38] to quantify the present-day biases of CESM2 with respect to Antarctic climate, atmospheric circulation, and AR frequency. MERRA-2 is the baseline for the Atmospheric River Tracking and Method Intercomparison Project (ARTMIP)[39,40], and has 3-hourly output and a horizontal grid spacing of 0.625˚ longitude by 0.5˚ latitude. This temporal and horizontal grid spacing is sufficient to resolve ARs and associated weather features and precipitation[15], and MERRA-2 compares best to ice core records of snow accumulation in Antarctica among multiple reanalyses[41]. From MERRA-2, we use vIVT to detect ARs, as well as IWV, 2 m temperature, 500 hPa geopotential height, mean sea level pressure, 500 hPa zonal winds, and 850 hPa meridional winds for comparisons to CESM2.

### AR detection
To identify ARs in present and future climates, we use a polar-specific AR detection tool (ARDT)[13]. The ARDT uses the 98th percentile of vIVT (defined as southward positive and calculated on a monthly basis at each grid point) from 1980 to 2014 as the threshold for AR detection between 39° S and 85° S. If contiguous filaments of vIVT exceed the aforementioned vIVT threshold and extend at least 20° latitude in length, they are classified as ARs. Here, vIVT is defined as:

$$v\text{IVT} = -\frac{1}{g} \int_{\text{surface}}^{\text{top}} qv \, dp \qquad (1)$$

with the standard gravitational acceleration, $g$ (m s$^{-2}$), specific humidity, $q$ (kg$^{-1}$ kg$^{-1}$), meridional wind speed, $v$ (m s$^{-1}$), and atmospheric pressure, $p$ (Pa).

This detection method is well-documented and heavily used for AR detection in Antarctica, as IVT-based AR detection tools (even those with lower thresholds for the polar regions) fail to capture ARs that penetrate into the interior of the ice sheet, due to the strong zonal winds around the

continent[20]. First, we apply the ARDT to CESM2, MERRA-2, and E3SMv2 for the present-day period, 1980–2014, to perform a comparison of AR frequencies between the two datasets. Due to the 20° latitude length requirement for ARs, spatial patterns in AR frequency at 60° S are highly sensitive to the northernmost latitude of the study domain at 39° S. Therefore, in order to perform a direct comparison between CESM2 and MERRA-2 between 1980–2014, we first regrid MERRA-2 vIVT to the CESM2 grid and then run the ARDT on both the regridded MERRA-2 and CESM2 data. For the bias comparison, we take the difference between the ensemble mean annual mean AR frequencies in CESM2 and the annual mean AR frequencies in MERRA-2 at each grid point. Regions with statistically significant differences are identified where the absolute value of the mean difference between the two datasets exceeds the standard deviation among CESM2 ensemble members.

Then, following the present-day analysis, we apply the same method of AR detection to CESM2 in the future climate from 2066–2100. We use the same monthly vIVT threshold as in the present-day, and if contiguous filaments of vIVT exceed the threshold and extend at least 20° latitude in length, they are classified as ARs.

### Scaling for atmospheric moisture increase

The vIVT threshold used to detect ARs is based on the present-day climatologies of atmospheric moisture and meridional winds over the Southern Ocean and Antarctica. However, future changes in these characteristics, particularly through the Clausius-Clapeyron effect, will influence the frequency of detected ARs. First, we examine the change in atmospheric moisture over the study region by analyzing the absolute and relative change in IWV at each grid point between 1980–2014 and 2066–2100, where regions with statistically significant differences are indicated where the difference in IWV between periods exceeds the standard deviation among CESM2 ensemble members. Then, for each grid point, we produce a time series of the relative monthly increase in IWV compared to the historical climatology for each ensemble member from 2015–2100. We determine the 10th percentile, mean, and 90th percentile of the relative IWV increase among the ensemble members for each month, to capture the range of variability in future moisture projections in CESM2. To generate smooth IWV scaling curves, we then take a 10-year running mean of the 10th percentile, ensemble mean, and 90th percentile curves and then calculate a third-degree polynomial trend line for each curve (Fig. 1c). These three curves determine the scaling mechanisms used to modify the AR detection threshold to account for increases in atmospheric moisture in the future climate, and increase over time from 2015–2100 in accordance with atmospheric moisture.

We apply each scaling level – the 10th percentile, ensemble mean, and 90th percentile trend line IWV ratios – to the present-day vIVT threshold for AR detection from 2066 to 2100 for each grid point, as follows. Here $ARthreshold_{future}$ represents the future threshold for ARs, $IWVratio_{future}$ represents the moisture scaling factor (10th percentile, ensemble mean, or 90th percentile relative increase in IWV), and $vIVTthreshold_{present}$ represents the present-day threshold of vIVT used to detect ARs.

$$\text{AR threshold}_{future} = \text{IWV ratio}_{future} * \text{vIVT threshold}_{present} \quad (2)$$

Thus, we develop four thresholds for future Antarctic AR detection using different levels to account for uncertainty in the increase in atmospheric moisture: (1) no scaling (the present-day vIVT threshold), (2) the ensemble 10th percentile of the relative increase in IWV among ensemble members (less restrictive than the mean increase), (3) the ensemble mean relative increase in IWV, and (4) the ensemble 90th percentile of the relative IWV increase (more restrictive than the mean increase). We then run the ARDT on CESM2 vIVT from 2066 to 2100 with the updated AR thresholds. For the presentation of results, we primarily focus on the difference between (1) no scaling and (3) scaling by the mean relative increase in IWV, though we provide frequency and precipitation analyses for all scenarios in the Supplementary Figs. For the frequency analysis presented in the main text, we compare AR frequencies between the future period (2066–2100) and the present-day (1980–2014), and mark regions with statistically significant differences wherever the absolute value of the mean difference in AR frequency between periods exceeds the standard deviation among CESM2 ensemble members.

### Calculating AR frequency

We use two metrics to quantify AR frequencies over the Antarctic Ice Sheet. First, we use the grid cell frequency of ARs to describe their occurrence at any given location over the Southern Ocean or the Antarctic continent. We calculate the grid cell frequency by summing the time steps with AR occurrence at each location in the domain and dividing by the total time steps. Second, we also describe the ice sheet-integrated annual frequency of ARs (including ice shelves). This value is calculated by counting the number of time steps per year there is an AR landfall anywhere over Antarctica (includes ice shelves, and time steps where multiple ARs make landfall at once) and dividing by the total number of time steps per year.

### Precipitation impacts attribution

From MERRA-2 reanalysis, we define total precipitation as the sum of snowfall and rainfall, where rainfall is the sum of convective and large-scale rainfall. Following version 5 of the Community Land Model, we partition CESM2 total precipitation into snowfall and rain over Antarctica using a linear ramp[42]. Precipitation that occurs with surface air temperatures below –2° C is classified as snowfall and precipitation that occurs with surface skin temperatures above 0 °C is classified as rainfall. For mixed precipitation occurring between −2 °C and 0 °C, we use a simple linear, temperature-based scale to partition precipitation into rain and snow (with $T_{sfc}$ representing the surface air temperature):

$$\text{rainfall} = \text{total precipitation} * \left( \frac{T_{sfc, -2°C \leq T \leq 0°C} + 2}{2} \right) \quad (3)$$

We focus on presenting total precipitation and rainfall, and not snowfall individually, because much of the rain falling in continental Antarctica is absorbed at the surface and refreezes, meaning that it contributes to the surface mass balance of the ice sheet[43]. We first calculate the mean annual total precipitation and rainfall over the Antarctic Ice Sheet from 1980–2014 for MERRA-2 and CESM2, and from 2066–2100 for CESM2. Next, we calculate AR-attributed total precipitation and rainfall over the present and future periods. Any precipitation that falls within the footprint of a detected AR mask at each time step is defined as AR-attributed precipitation[9]. Furthermore, any precipitation that falls within the footprint of an AR mask up to 24 hours after the AR has occurred is attributed to the AR. We apply this attribution method to CESM2 and MERRA-2 reanalysis for the present-day (1980–2014). For the future period (2066–2100), we calculate AR-attributed precipitation in CESM2 using the four AR catalogs generated in subsection "Scaling for atmospheric moisture increase" of section "Methods" (no scaling, 10th percentile scaling, mean scaling, and 90th percentile scaling). While the IWV-scaled catalogs serve to test sensitivity of AR-attributed precipitation to the AR detection threshold, the no scaling scenario (based just on the present vIVT threshold) serves as a baseline for evaluating future changes in AR impacts relative to the present-day climatology.

In addition to calculating the mean annual total precipitation and rainfall as well as the mean annual AR-attributed total precipitation and rainfall in present and future climates, we also examine the relative contribution of ARs to all precipitation and their contribution to year-to-year variability in precipitation. To calculate the relative contribution of ARs to the total precipitation at each grid cell, we take the annual AR-attributed precipitation and divide by all total precipitation that year, then take the annual mean over the present and future climatological periods, respectively (1980–2014 and 2066–2100). An identical method is used for rainfall. For CESM2, we then take the mean of the relative contributions calculated amongst each of the 40 ensemble members. To calculate the contribution of

**Article**

ARs to year-to-year variability in precipitation (or the percent variance explained), we first detrend a time series of the AR precipitation and all precipitation at each grid cell by subtracting the linear trend from each time series. Then, we calculate the Pearson correlation coefficient between detrended AR-attributed precipitation and detrended total precipitation at each grid cell (and the same for rainfall). By taking the square of the correlation coefficient, we determine the percent variance of precipitation explained by ARs.

To analyze the bias of present-day total precipitation and rainfall in CESM2 relative to MERRA-2, we regrid MERRA-2 precipitation to the CESM2 grid, take the difference, and mark areas where the absolute value of the mean difference between the CESM2 and MERRA2 exceeds the standard deviation among CESM2 ensemble members. For each comparison that presents differences between future precipitation (total precipitation and rainfall) and present-day precipitation, we similarly highlight regions of statistical significance wherever the absolute value ensemble mean difference between periods exceeds the standard deviation among CESM2 ensemble members. For figures showing the relative contribution of ARs to year-to-year variability in precipitation, we indicate areas where the contribution of ARs is statistically significant based on a p-value less than 0.1. For figures showing future changes in the relative contribution of ARs to total precipitation, only regions meeting both requirements described above are marked as statistically significant.

## Circulation changes

To examine present-day atmospheric circulation biases and future changes in CESM2, we isolate a series of circulation variables by season. First, we analyze the bias in CESM2 relative to MERRA-2. For each variable (500 hPa geopotential height, 500 hPa zonal winds, 850 hPa meridional winds, and mean sea level pressure), we identify the seasonal mean in MERRA-2 and CESM2 from 1980–2014. We then take the difference between CESM2 and MERRA-2, and mark regions with statistically significant differences wherever the absolute value of the difference between ensemble mean CESM2 values and MERRA-2 values exceeds the standard deviation among CESM2 ensemble members. Then, we take the difference between the seasonal mean in the future period and the present period for each variable in CESM2. Here, we perform a two-tailed z-statistic for each circulation variable to determine significantly different means between the two periods at the 95% confidence interval (critical value of 1.96). We apply the same method to quantify future changes in AR frequency and plots these changes alongside the set of circulation variables. Finally, we also examine the difference in sea ice fraction between the future period and the present-day, marking significant regions wherever the absolute value of the ensemble mean difference between periods exceeds the standard deviation among CESM2 ensemble members.

## Data availability

The Wille AR catalog using MERRA-2 reanalysis, as well as CESM2 (CAM5) source data, can be found on the Climate Data Gateway website, https://rda.ucar.edu/datasets/d651012/. The CESM2 AR catalogs generated as a part of this study are hosted within the University of Colorado Boulder CU Scholar archive at the following links: https://scholar.colorado.edu/concern/datasets/k930bz575 (1980–2014), https://scholar.colorado.edu/concern/datasets/9306t087b (2015–2100 with no scaling), https://scholar.colorado.edu/concern/datasets/jh343v05h (2015–2100 with 10th percentile IWV scaling), https://scholar.colorado.edu/concern/datasets/9k41zg35v (2015–2100 with mean scaling), https://scholar.colorado.edu/concern/datasets/3j333378z and (2015–2100 with 90th percentile IWV scaling). E3SMv2 code and data are published under the E3SM Project as Energy Exascale Earth System Model (E3SM), October 11, 2021 release (version 2.0), at https://doi.org/10.11578/E3SM/dc.20240301.3.

## Code availability

The code used for AR detection in this study, developed by J.D.W. is available at https://github.com/jonathanwille/Antarctic-lab.

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

## Acknowledgements

M.L.M. and A.C.W. were supported by National Aeronautics and Space Administration grant no. 80NSSC21K1610. C.A.S. was supported by the Regional and Global Model Analysis (RGMA) component of the Earth and Environmental System Modeling Program of the U.S. Department of Energy's Office of Biological and Environmental Research (BER) under Award Number DE-SC0022070 and the National Center for Atmospheric Research, which is a major facility sponsored by the National Science Foundation (NSF) under Cooperative Agreement No. 1852977. R.T. was supported by National NSF Award OPP-2043727. J.D.W. was supported by Horizon 2020 project nextGEMS under grant agreement number 101003470.

## Author contributions

M.L.M. generated the conceptual direction of the work, carried out the analysis, and produced the figures and manuscript. A.C.W., C.A.S., R.T., L.B., F.C., and J.D.W. contributed to the conceptual direction of the work. A.C.W. edited the manuscript, CAS contributed to the analysis and writing, and J.D.W. contributed to the analysis.

## Competing interests

The authors declare the following competing interest: C.A.S. is an associate editor for NPJ Climate and Atmospheric Sciences. All other authors declare no competing interests.

## Ethics approval and consent to participate

This study was conducted in accordance with the principles of inclusivity and ethical research practices.
