## [Transparent Peer Review file · Communications Earth & Environment]

Rising atmospheric moisture escalates the future impact of atmospheric rivers in the Antarctic climate system

Corresponding Author: Dr Michelle Maclennan

Version 0:

Decision Letter:

Dear Dr Maclennan,

Your manuscript titled "Rising atmospheric moisture escalates the future impact of atmospheric rivers in the Antarctic climate system" has now been seen by 4 reviewers, whose comments are appended below. You will see that they find your work of some potential interest. However, they have raised quite substantial concerns that must be addressed. In light of these comments, we cannot accept the manuscript for publication, but would be interested in considering a revised version that fully addresses these serious concerns. In addition, please consider the following editorial thresholds:

1. Fully justify model and method choices by addressing CESM2's weakness, the resolution choice for AR detection, and the selection of CESM2 LE members over other models.
2. Clarify comparisons and outputs by improving the comparison between model projections and MERRA-2 data, particularly regarding IWV changes, and better explaining the relationship between sea ice loss and poleward flow, as well as the jet shift.
3. Consider expanding on climate impacts by highlighting how extreme ARs will affect Antarctica's surface mass balance, with a focus on snowfall, rainfall, and how ARs could increase surface melt and runoff.

We hope you will find the reviewers' comments useful as you decide how to proceed. Should additional work allow you to address these criticisms, we would be happy to look at a substantially revised manuscript. If you choose to take up this option, please either highlight all changes in the manuscript text file, or provide a list of the changes to the manuscript with your responses to the reviewers.

When resubmitting, please provide a point-by-point response to the reviewers' comments. Please submit your responses as a separate file, distinct from your cover letter where you can add responses to the Editors' comments that you do not want to be made available to the reviewers. Word files are preferred. We recommend that any figures, tables or graphs that are included in the response to reviewers are also included in the main article or Supplementary Information.

If the revision process takes significantly longer than three months, we will be happy to reconsider your paper at a later date, as long as nothing similar has been accepted for publication at Communications Earth & Environment or published elsewhere in the meantime.

Please use the following link to submit your revised manuscript, point-by-point response to the reviewers' comments with a list of your changes to the manuscript text (which should be in a separate document to any cover letter), a tracked-changes

version of the manuscript (as a PDF file) and any completed checklist:

Link Redacted

Please do not hesitate to contact us if you have any questions or would like to discuss the required revisions further. Thank you for the opportunity to review your work.

Best regards,

Jose Luis Iriarte Machuca, PhD
Editorial Board Member
Communications Earth & Environment

Alireza Bahadori, PhD
Associate Editor
Communications Earth & Environment

EDITORIAL POLICIES AND FORMAT

If you decide to resubmit your paper, please ensure that your manuscript complies with our editorial policies and complete and upload the checklist below as a Related Manuscript file type with the revised article:

Editorial Policy Policy requirements
(Download the link to your computer as a PDF.)

- Behavioural and social science
- Ecological, evolutionary & environmental sciences
- Life sciences

<https://www.nature.com/documents/nr-reporting-summary.zip>

For your information, you can find some guidance regarding format requirements summarized on the following checklist: (<https://www.nature.com/documents/commsj-phys-style-formatting-checklist-article.pdf>) and formatting guide (<https://www.nature.com/documents/commsj-phys-style-formatting-guide-accept.pdf>).

REVIEWER COMMENTS:

Reviewer #1 (Remarks to the Author):

Review of Maclennan et al. "Rising atmospheric moisture escalates the future impact of atmospheric rivers in the Antarctic climate system"

Motivated by the key role the Antarctic Ice Sheet plays in the modulation of present and future global mean sea levels and by the role that precipitation events (in particular synoptic-scale extreme events) play in compensating mass loss from Antarctica to the ocean each year, the authors set out to investigate future changes in Antarctic atmospheric river events and characterize the role they play in the future mass balance of the Antarctic Ice Sheet.

Through the use of the CESM2 large ensemble under the ssp3-7.0 scenario, the authors show that there is projected to be increase in the integrated water vapor (IWV) over the Antarctic continent that represents the largest relative change in IWV at Southern Hemisphere high latitudes. The authors further show that, in addition to the increase in IWV, there is an increase in poleward integrated vapor transport (vIVT) at Southern Hemisphere high latitudes. Through the use of a polar-specific atmospheric river detection tool, the authors show that there is projected to be an increase in the atmospheric river days per year as a result of the projected changes in IWV and vIVT. To separate thermodynamic and dynamic effects, the authors scale the atmospheric river detection threshold based on the increased IWV, and they find that shifting climatological regions preferential to cyclogenesis lead to smaller increases and decreases in atmospheric river frequency similar to a zonal wave three pattern. Further, the authors note that using this scaling for future atmospheric river detection plays a large role in quantifying the relative role that extreme events like atmospheric rivers play in the overall mass balance and Antarctic contribution to global mean sea level rise.

I find the scope of the work is appropriate for the journal and the analysis of projected changes in Antarctic atmospheric

rivers as a result of thermodynamic and dynamic changes to be a worthwhile addition to previous work that has also suggested that future changes in atmospheric rivers are highly sensitive to their detection methods. I do have a few comments that may help clarify some aspects of the manuscript, but as I do not expect my comments to significantly impact the main conclusions of the paper, I recommend minor revisions.

Minor comments:

1. I appreciate that the authors include MERRA-2 as a baseline to understand the bias of the CESM2 ensemble, but it piqued my curiosity about whether the changes the authors project in the future climate are visible in observation-based data as well (ie, can you see the increase in IWV as in Fig. 1c but for MERRA-2 data, or is there a trend in AR days per year in MERRA2 based on the current threshold of vIVT for AR detection as in Fig 2c), or is interannual variability too large to get these trends without the use of the ensemble?
2. Section 2.2: Would it be worthwhile to scale each ensemble member based on its own change in IWV (the individual blue lines in Fig. 1c) opposed to the curves based on the mean, 10th, and 90th percentiles, or is this too noisy due to interannual variability?
3. Section 2.4: I found this section to be quite interesting! However, I was thrown off at first by the description of the projected jet shifts in the Southern Hemisphere as a result of climate change to be 'eastward,' as previous literature (e.g. Kushner et al. 2001; Barnes and Polvani 2013, among others) typically describe the jet response to climate change to be poleward when taken in the zonal mean. After consulting the supplemental material (in particular S13b2), I understand what is meant by an eastward shift in the manuscript as in addition to the well-known polar shift (blue shading poleward of the jet), it appears that the location of the jet max also shifts eastward based on the contours. Perhaps in order to avoid any confusion, it may be preferable to denote this shift as a shift in the maximum jet winds, or a shift in the jet streak, climatologically.
4. Lines 276-281: While areas of increased poleward 850 flow are co-located with a reduction in sea ice offshore, it appears to me from S15 that reductions in sea ice offshore are present at pretty much all longitudes regardless of changes in poleward flow. Is there relatively more sea ice loss in the regions identified in the manuscript?
5. Section 4.5: This will likely have no qualitative impact on the manuscript, but as you use a sample of 40 CESM2 LE members (and, truth be told, the 100 total members can be thought of as a sample of all possible realizations, rather than the full population of CESM2 realizations) a t-statistic may be more appropriate to use in place of a z-statistic, which has a slightly greater critical value for a two-tailed 95% confidence interval with 39 degrees of freedom.

Other general comments:

1. While the manuscript is well-written and clear, there are a couple instances (and I may have missed others) where it looks like words were accidentally omitted or retained in the writing process. Examples of this I noticed were on Line 51, 78-79, and the caption of Fig. 2.
2. While not necessary, the authors may consider editing the labels in the supplemental material as in some figures there are multiple sets of panels labeled a,b,c, and d, some of which get mentioned in the main text of the manuscript.

References

1. Barnes, E. A. & Polvani, L. Response of the Midlatitude Jets, and of Their Variability, to Increased Greenhouse Gases in the CMIP5 Models. *Journal of Climate* 26, 7117–7135 (2013).
2. Kushner, P. J., Held, I. M. & Delworth, T. L. Southern Hemisphere Atmospheric Circulation Response to Global Warming. *J. Climate* 14, 2238–2249 (2001).

Reviewer #2 (Remarks to the Author):

Please, refer to attached File

Reviewer #3 (Remarks to the Author):

Review of Maclennen et al 2024: Rising atmospheric moisture escalates the future impact of atmospheric rivers in the Antarctic climate system

Review summary

This article addresses how atmospheric rivers (ARs) may change in Antarctica under future climate conditions, using an ensemble of CESM2 simulations under the SSP3-7.0 scenario. The findings highlight two possible outcomes: if ARs are detected using present-day thresholds, their frequency increases significantly in the future. Conversely, when thresholds are scaled to account for rising atmospheric water vapor due to warming, AR frequency shows only moderate increases or even slight regional declines due to changes in atmospheric circulation. While this result may seem inconclusive to some, the study offers a critical contribution to understanding ARs and their role in Antarctic surface and total ice sheet mass balance. The study projects substantial increases in both snowfall and rainfall over Antarctica as the climate warms. However, attributing these changes directly to ARs depends on the definition of what constitutes an AR.

Overall, I found the manuscript to be methodologically sound, well-written, and supported by clear figures in the main text and supplementary information. The authors provide a careful, nuanced discussion of how ARs and their impacts may

evolve, comparing CESM2 with the MERRA-2 reanalysis for present-day conditions and using ensemble-based simulations to explore uncertainty in future projections. This paper addresses an important topic with relevance to considering the future evolution of the Antarctic ice sheet and its contribution to sea level. I look forward to seeing this article published and I expect that many others in the Antarctic and climate science community would feel the same.

While the paper is strong, there are some areas where the authors could provide additional context and emphasis to enhance the paper's clarity and impact. I thank the authors in advance for considering my comments.

Sincerely,
Luke Trusel

Broad comments:

I would encourage the authors to consider whether they may be underselling their analysis. The authors might consider emphasizing that extreme events considered rare today will become far more common under future warming. This framing could help underscore the importance of ARs on Antarctica's surface mass balance (SMB). Similarly, greater focus could be placed on the projected increases in both snowfall and rainfall. While rainfall is a smaller component of total precipitation, its relevance for ice shelf stability and the liquid water budget could be more explicitly discussed.

The focus of the paper is on the SMB response (i.e., snowfall), yet these ARs will also be driving large increases in surface melt (and the liquid water budget overall via rainfall). While the authors do report changes in rainfall, there's no discussion of how melt may change in the future. I understand CESM2 may not represent surface melt directly, but the authors could assess temperature metrics as a proxy for melt. I think equally important to whether ARs will increase in the future is the question of how their impacts will change: will they be mainly positive by delivering enhanced snowfall, or will they drive substantial increases in melt and runoff that may lead to far reduced SMB? I would encourage the authors to give this some thought and consider how explicit analysis of melt or discussion of it might be incorporated in the manuscript.

Specific comments:

Line 99: Figure 1c effectively highlights uncertainty in future moisture changes. However, other figures (e.g., stippling in Figure 1a) suggest lower ensemble spread, as almost all areas are stippled. A higher threshold for statistical significance (e.g., 2 standard deviations) might better represent regions of true uncertainty.

Line 114: Consider adding "in these regions" to the end of the sentence for clarity

Line 117: Please clarify the discrepancy between the 15% AR frequency reported here and the 1% frequency mentioned in the introduction. Figure S5d also suggests ~3% frequency without IWV ratio scaling, which adds to my confusion.

Line 134: Probably good to cite Raphael 2004 with respect to zonal wave three:
Raphael, M. N., 2004: A zonal wave 3 index for the Southern Hemisphere. *Geophys. Res. Lett.*, 31, L23212,
<https://doi.org/10.1029/2004GL020365>.

Figure 2 caption: Correct the text: "scaled by thees relative average relative" to "scaled by these relative average changes in IWV." Also, explicitly reference Figure 1b if relevant.

Lines ~156-160: The authors remain neutral on whether AR detection thresholds should be scaled for future IWV increases. Adding recommendations or arguments for and against scaling would be helpful. For instance, if the goal is to classify ARs consistently under changing climates, scaling might be necessary.

Line 174/195: I am concerned over the large magnitude of rainfall in present-day CESM2, especially on the Antarctic Peninsula (Figure S6). I think more discussion of this bias is warranted. Given that ARs are also relevant to the liquid water budget (especially in the future), how this bias might impact projections of future SMB change or ice shelf instability is warranted. Line 195 somewhat downplays the rainfall bias as being a small component of the total precipitation. Yes, this is true, but rainfall here is a large fraction of the liquid water budget (e.g., melt rates today on the northern AP are ~400 mm w.e./yr and CESM2 suggests ~200 mm w.e./yr rainfall here in the present-day).

Section 2.4 / Line 212 / Line 223: There's some discussion of sea ice here, but not much context for its relevance. Is the sea ice responding to the changes in atmospheric circulation? Are ARs in the model causing reduced sea ice? Is the reduced sea ice leading to more moisture? Some context would be helpful.

Line 220 (and elsewhere regarding polar jet): Please clarify how changes in geopotential height reflect the eastward shift of the polar jet. Additional explanation of the methods used to determine jet stream changes would improve clarity.

Line 231: Discussion here how there's an increase in zonal winds and this impacts the detection of ARs using the (meridional) vIVT method. It could be mentioned that other ARDTs might produce better classifications (e.g., ones based on IWV or uIVT).

Reviewer #4 (Remarks to the Author):

The manuscript "Rising atmospheric moisture escalates the future impact of atmospheric rivers in the Antarctic climate system" by Michelle Maclennan et al. analyses the sensitivity of Antarctic atmospheric rivers (ARs) to future changes in atmospheric moisture in terms of AR frequency and AR-related precipitation. For this purpose, they compare the present-day climate (1980 – 2014) and the future climate (2066 – 2100) based on the MERRA-2 reanalysis product and the global climate model CESM2 LE. They generally find an increase in AR frequency and precipitation impacts; however, these results are sensitive based on the detection algorithm and the water vapor increase. Finally, they analyze the linkage between circulation changes and atmospheric river activity.

Since the ARs play an important role in the Antarctic Ice Sheet, it is of particular interest to investigate how the AR occurrence and the AR-related precipitation will change in the future. Therefore, this study gives a good overview of future changes and sensitivity in AR-frequency and AR-related precipitation in the Antarctic.

The manuscript is well written, and I think it is a valuable study for the AR community and the analysis of future changes in Antarctica. Further, it is helpful to better assess the changes in the Antarctic ice sheet related to AR precipitation. However, I have some questions about the methodology and suggestions for improving the structure and figures which you can find in the attached document.

Communications Earth & Environment is committed to improving transparency in authorship. As part of our efforts in this direction, we are now requesting that all authors identified as 'corresponding author' create and link their Open Researcher and Contributor Identifier (ORCID) with their account on the Manuscript Tracking System prior to acceptance. ORCID helps the scientific community achieve unambiguous attribution of all scholarly contributions. You can create and link your ORCID from the home page of the Manuscript Tracking System by clicking on 'Modify my Springer Nature account' and following the instructions in the link below. Please also inform all co-authors that they can add their ORCIDs to their accounts and that they must do so prior to acceptance.

Version 1:

Decision Letter:

Dear Dr Maclennan,

Your revised manuscript titled "Rising atmospheric moisture escalates the future impact of atmospheric rivers in the Antarctic climate system" has now been seen by our reviewers, whose comments appear below. In light of their advice we are delighted to say that we are happy, in principle, to publish a suitably revised version in Communications Earth & Environment.

We therefore invite you to revise your paper one last time to address the remaining concerns of our reviewers. At the same time we ask that you edit your manuscript to comply with our format requirements and to maximise the accessibility and therefore the impact of your work.

EDITORIAL REQUESTS:

****Please take care to match our formatting and policy requirements. We will check revised manuscript and return manuscripts that do not comply. Such requests will lead to delays. ****

SUBMISSION INFORMATION:

OPEN ACCESS:

Communications Earth & Environment is a fully open access journal. Articles are made freely accessible on publication. For further information about article processing charges, open access funding, and advice and support from Nature Research, please visit <https://www.nature.com/commsenv/open-access>

Link Redacted

Best regards,

Jose Luis Iriarte Machuca, PhD
Editorial Board Member
Communications Earth & Environment

Alireza Bahadori, PhD
Associate Editor
Communications Earth & Environment
Consulting Editor
Communications Sustainability

REVIEWERS' COMMENTS:

Reviewer #1 (Remarks to the Author):

I appreciate the lengths the authors have gone to address my concerns, and the edits they have made in relation to my comments, as well as those by other reviewers. In particular, I find the edits to section 2.4 and elsewhere have greatly clarified the nature of the circulation changes and how they relate to AR activity in the Antarctic.

I offer one small clarification regarding one of my previous comments, however I appreciate that the authors tested whether the choice of statistical test was important or not (and as expected, it had no qualitative impact on the nature of the results). My main point was when commenting on the choice of a z- or t-test was that even if all members of the ensemble are used, that does not necessarily represent the true population statistics of the model in question. Again, though, I greatly appreciate the lengths the authors went to within their response to confirm that the results of their analysis were not sensitive to the choice of a z-test or a t-test with 39 degrees of freedom.

As the authors have sufficiently addressed my minor concerns, I find the work to be a worthwhile addition to the journal.

Reviewer #2 (Remarks to the Author):

I believe that the authors have correctly addressed most of my comments. I congratulate them on their work. Now I feel comfortable giving my recommendation for publication.

Reviewer #3 (Remarks to the Author):

I have reviewed the authors' responses to my previous comments and am satisfied with the revised manuscript. This is an important paper and I look forward to seeing it published!

Reviewer #4 (Remarks to the Author):

In this study, the authors assess the sensitivity of Antarctic atmospheric rivers (ARs) to projected changes in atmospheric moisture, focusing on both AR frequency and AR-associated precipitation (and rainfall). For their analysis, they use the MERRA-2 reanalysis product (for the current climate) and the CESM2 LE global climate model (for the current and future climate) under the SSP3.7 scenario. In general, the authors find an increase in both AR frequency and precipitation impacts. However, the results are sensitive to the detection algorithm and the magnitude of the water vapor change. Using today's threshold, AR frequency will increase significantly in the future. In contrast, a higher threshold due to warming and hence higher moisture content shows only a moderate increase. The study concludes by linking shifts in atmospheric circulation to variations in AR activity.

Overall, I found the manuscript well written and the methodology clear. The study provides valuable insights for the AR research community, especially in the context of assessing future changes in Antarctica. However, the results should also be treated with caution because only one climate model was used and, as the authors also write, there are uncertainties in the AR detection algorithm. In this case, I would have liked to see other climate models used. Apart from that, I have three minor comments:

L100: -1? Is the minus correct?

L120 and in the Supplement Figure S3: Is the AR frequency in % or days per year?

L190: up to 30% of the total: Do you mean rainfall? If yes, I would include rainfall – it becomes more clear that you don't mean total precipitation

Rising atmospheric moisture escalates the future impact of atmospheric rivers in the Antarctic climate system

Maclennan et al.,

Proposed for publication in Nature Com. Earth & Environment

Round 1 · Dec 2024

Reviewer Report

This manuscript examines how atmospheric rivers will affect surface mass balance in Antarctica under future climate change scenarios (SSP3-7.0). Using a set of 40 Community Earth System Model 2 (CESM2) simulations, the authors project that the frequency of ARs could double and their contribution to Antarctic precipitation could triple by the end of the 21st century, using current detection thresholds.

The authors claim that their results show a high sensitivity of projections to AR detection methods, highlighting the need for moisture-based scales for future simulations. The study identifies increased precipitation driven by ARs in coastal regions, particularly in West Antarctica, and highlights the potential interaction of these events with cryospheric and oceanic systems, such as retreating anchor lines in glaciers.

Overall, the paper is very well written and provides interesting results, especially considering the importance of the study region and the scarcity of similar research. However, I am not sure that it reaches the right level for such a high impact journal, and there are also some issues that concern me that I would like the authors to discuss before giving my recommendation for publication.

Major Comments

1. Although the authors use robust ensembles, they limit their study to the results of a single model, the CESM2. This choice cannot be criticized and is as good a choice as any other. In any case, CESM2 has a number of weaknesses (e.g. high climate sensitivity) that I think should at least be discussed in the manuscript. Of all these, in this case (because of the "similarity" between the two regions) I think the "poor" ability it has shown to resolve the Arctic summer ice cover (see <https://doi.org/10.1029/2020jc016133>) is particularly relevant. I am not too concerned in this case, but I would like to know if the authors have taken this into account; and perhaps include a small discussion of it in the manuscript.

2. I understand the limitations of the data sources, but I need a discussion on this: is $1.25^\circ \times 0.95^\circ$ an adequate resolution for AR detection? Experience tells me that this is a dangerously low resolution, so I ask the authors to discuss whether its use is justified. It would also not hurt to add a few sentences discussing the pros and cons of ARDT. In particular, I was surprised that the detection method focuses only on the meridional flux of water vapor. Perhaps this is justified in the region of interest, but it is not the norm.

Minor Comments

Introduction: I fully understand the space constraints of this journal, and the introduction is well done (and coherent). However, I think some relevant bibliography on the role played by ARs in Antarctica is missing and could be cited. For example, take a look at:

<https://doi.org/10.1002/wcc.588>

<https://doi.org/10.3390/w11010041>

L60: I would not say that ARs are the most intense mechanism of moisture transport in the atmosphere. In fact, they are not; they are the major advective mechanism of meridional water vapor transport between the extratropics and mid and high latitudes, which is not the same thing. Also, the commonly used acronym is "ARs", not "ARS".

L74: I congratulate the authors for choosing the SSP370 scenario, which is certainly much more likely and reasonable than SSP585, which for some reason we tend to use heavily in the community. In any case, please include a brief justification for its use in the manuscript.

Fig. 3 Wouldn't it be more logical to put the figure upright instead of on its left side? It is not comfortable for the reader to read vertical text...

Fig. 3, caption. Last line, p-value < 0.1 ??

L253: Sorry, but how exactly do your results show that? Have different detection methods been applied?

The manuscript “Rising atmospheric moisture escalates the future impact of atmospheric rivers in the Antarctic climate system” by Michelle Maclennan et al. analyses the sensitivity of Antarctic atmospheric rivers (ARs) to future changes in atmospheric moisture in terms of AR frequency and AR-related precipitation. For this purpose, they compare the present-day climate (1980 – 2014) and the future climate (2066 – 2100) based on the MERRA-2 reanalysis product and the global climate model CESM2 LE. They generally find an increase in AR frequency and precipitation impacts; however, these results are sensitive based on the detection algorithm and the water vapor increase. Finally, they analyze the linkage between circulation changes and atmospheric river activity.

Since the ARs play an important role in the Antarctic Ice Sheet, it is of particular interest to investigate how the AR occurrence and the AR-related precipitation will change in the future. Therefore, this study gives a good overview of future changes and sensitivity in AR-frequency and AR-related precipitation in the Antarctic.

The manuscript is well written, and I think it is a valuable study for the AR community and the analysis of future changes in Antarctica. Further, it is helpful to better assess the changes in the Antarctic ice sheet related to AR precipitation. However, I have some questions about the methodology and suggestions for improving the structure and figures.

Specific comments:

- In the last sentence of the abstract (LL 33 – 36), you mention the importance of using large ensembles compared to few ensemble members. This statement sounds logical. However, I cannot find analyses that compare large ensemble members to a few.
- Can you better explain the calculation of the IWV ratio in equation (2)? It becomes more clear in the caption of Figure 1c) but here, the IWV ratio varies for each year. Thus, it is still unclear which value you used for the $IWV_{ratio\ future}$.
- For the future climate, you use the CESM2 LE. Why are you using this model? Is there a specific reason? For the model uncertainty, you compare it with E3SMv2. Would it make more sense to use a multi-model ensemble? How did you choose the 21 ensemble members for CESM for the comparison with E3SMv2?
- For the analysis, you use 40 ensemble members from CESM2 LE. Initially, I wondered why you chose these 40 ensemble members. Later, you mention that you use the ensemble members with smoothed biomass burning. I think you should mention this earlier. Can you also explain why you chose these ensemble members?

- I am not sure if it would be better to swap chapters 2.3 and 2.4. I think chapter 2.4 is more related to 2.2.
- Comments to figures:
 - Figure 1a): It might be better to use a colormap with more contrast.
 - Figure 1c): It is difficult to see the mean trend line. I would change the color for the ensemble mean or the mean trend line.
 - It is a bit confusing that you are not consistent with the colormap. For example, in Fig. 2, negative values are red, and positive values are blue in Figs. S3, S11, and S12, it is reversed.
 - Interestingly, you have a significant change for all grid points in Fig. 1a) and b), Fig. 2c), and also in the supplementary.
 - Supplementary: For most images, you distinguish between left and right and use a-d for each image. I would prefer the letters a-h. I think that would make it a bit clearer.
 - I like Figure 5: It gives a very good overview!

Minor comments:

- L29: In the discussion (Chapter 3 LL 248-249), the increase in precip is three times higher
- L30: which surface impacts? Only precip? At least you apply the algorithm only to precipitation. The changes in other surface variables (500 hPa geopotential height, 2m temperature, 500 hPa zonal wind...) are not related to the threshold.
- L61: ARS → AR_s
- L78: present day → present-day
- LL94-95: I would relocate Fig. S2 after the Antarctic continent in L94. In this position, I would expect a temperature map in Fig. S2.
- L103: Here you write: CESM2 large ensemble. Sometimes, you write CESM2 or CESM2 LE → Is CESM2 LE all ensembles and CESM2 only the 40 ensemble members?
- L109: the low-pressure bias is only seen in Fig. S3; Fig. S4 shows the difference between CESM and E3SMv2 → mention Fig. S4 earlier in L108 in addition to Fig. 2b
- L112-114: Do you think this blocking system is responsible for an AR increase over the Ross Sea?
- L121: present day → present-day
- L136: (Gt) → Gt

- L140: ARs explain 3—50% of the interannual variability → Change colormap in Fig. 3c, maximum 50%
- L145: Fig. S6 → (Fig. S6)
- LL147-148: change colormap from -10 to 10%
- LL152-153: Increases in AR precipitation outpace increases in total precipitation → cannot see this in Fig. S7. Do you mean Fig. S8
- LL 154: 24 +/- 2% in Fig. 3e → adapt colormap
- L160: 'No change'? → There is a change but not so high – maybe more precise: there is no significant change?
- LL164-167: In which Figure can you see this?
- LL174-175: change colormap in Fig. 4a,b → especially on the coastal lines, it is difficult to distinguish between dark blue and black
- L183-184: I cannot see the AR-related rainfall in Fig. S7. Do you mean Fig. S9?
- L185: The maximum value in Fig. 4e is 24%. Do you mean 21-24%?
- LL259-262: Is this the reason that you use CESM2?
- L282: present day → present-day
- L322-323: Why do you know that this resolution is sufficient to resolve ARs? → Is there a reference?
- LL323-325: I cannot find a comparison between MERRA2 and ice core records in this paper
- LL349, 351, 354, 372, 376, 385, 408, 431, 436, 445: present day → present-day
- L362: time series for each grid point, or only for significant grid points?
- LL362-364: Rewrite sentence: Then, for each grid point, we produce a time series of the relative monthly increase in IWV for each ensemble member from 2015 to 2100 compared to the historical climatology
- L372: I don't understand the calculation of the IWV ratios
- L398-400: I don't understand this statement. It sounds like you present total precipitation only – but later, you compare between total precipitation and rainfall.
- L401-404: Have you regrid MERRA-2 precip to CESM2 grid? → Yes, you mention it in LL 431-432 → mention it earlier
- L450: Have you regrid MERRA-2?
- L459: between the future period and the present day → difference for CESM2 only?
- Caption Fig. 2 LL4-5; Caption Fig. 3 L3; Caption Fig. 4 L2: present day → present-day
- Eq. 3: include T_{sfc} between -2 and 0C
- Figure 1:
 - It seems that the difference is significant for all grid points. Is this true?
- Figure 2:
 - Caption L2: present → present-day climate

- Caption L6: scaled by these relative average relative increases in IWV → I don't understand
- Figure 3 and Figure 4:
 - Caption: After (a), (b), (c), (d and g) begin sentence with capital letter
 - Do you mean $p < 0.1$?

Response to reviewer comments on “Rising atmospheric moisture escalates the future impact of atmospheric rivers in the Antarctic climate system”

Michelle L. Maclennan, Andrew C. Winters, Christine A. Shields, Rudradutt Thaker, Léonard Barthelemy, Francis Codron, and Jonathan D. Wille

Dear Reviewers,

We would like to thank you for your assessment of our manuscript, and for the constructive feedback provided. Here, we submit a revised version of the original manuscript, addressing all of the comments raised by reviewers. In particular, we there were primary areas for revision: (1) justification for the use of the CESM2 large ensemble, (2) clarification of present-day trends in MERRA-2, biases in CESM2, and how these relate to future changes, and (3) a greater emphasis on the relevance of increased AR rainfall in the context of Antarctic surface mass balance and surface melt.

Based on reviewer feedback, we recognize the importance of more fully justifying the choice of the CESM2 large ensemble (and the 40 smoothed biomass burning members) for this study. We use the CESM2 large ensemble because it enables us to examine ARs in the present and future climate (up to 2100) in 40 ensemble members at 6-hourly temporal resolution with 32 vertical levels, including all of the variables we required for this study (vIVT, IWV, 2m temperature, sea ice fraction, total precipitation, surface skin temperature, 500 hPa geopotential height, mean sea level pressure, 500 hPa zonal wind, and 850 hPa meridional wind). We added justification for the 1-degree grid spacing of the model regarding its representation of ARs while acknowledging the advantage of finer-resolution products, such as regional climate models, which would provide higher-fidelity representations of AR structure, interaction with Antarctic topography, and surface impacts. We also added context to the Datasets section on our choice of the SSP3-7.0 radiative forcing scenario because of its importance as a realistic, middle-high end emissions scenario. Although applying a full CMIP6 multi-model ensemble was beyond the scope of this work, we address model uncertainty by comparing two different models, with different ocean components, different model biases, and different variability and climate change responses.

Throughout the manuscript, we increased the discussion of how present-day biases in CESM2 relative to MERRA-2 impact our future projections for Antarctic ARs and their impacts. We clarified the role of future changes in IWV (in particular, the uniformly significant increase in IWV relative to the ensemble spread) and performed an assessment of the present-day trends and variability of ARs in MERRA-2 reanalysis. We also strengthened our discussion of the relationship between sea ice, poleward flow, and CESM2 biases in Antarctic precipitation. In particular, we connected negative sea ice area biases in CESM2 to the positive biases in Antarctic precipitation, and addressed how these may perpetuate future overestimates of AR contributions to Antarctic precipitation. Finally, we heard from two reviewers that our description of the eastward shift in the polar jet in future climates was unclear. In response, we refined the

language to more explicitly describe the change in 500 hPa zonal wind speeds as an "eastward shift in the maximum wind speed of the polar jet", and connected this change closely with the present-day bias in the positioning of the polar jet in CESM2.

Reviewer #3 in particular called for more detail on the broader surface impacts of future ARs in the context of both surface mass balance and ice shelf stability. We recognize the value of contextualizing our results on AR-attributed precipitation, particularly rainfall, and added details throughout the manuscript to address this comment. In the Introduction, we added a sentence describing AR impacts on surface melting and ice shelves in the present-day, whereas in the original manuscript we had focused on present-day precipitation impacts. In the Results section, we strengthened our language on the magnitude of future increases in AR frequency and precipitation, to underscore how these rare, extreme events become much more common in future climates. We expanded our discussion of increases in AR rainfall - addressing the regionality of the rainfall increase (highest increases on the Antarctic Peninsula) and the adverse impacts it may have on ice shelf stability. To provide more nuance to the rainfall discussion, we added a discussion of how the strong present-day biases in CESM2 rainfall may inflate future projections over the Antarctic Peninsula. Ultimately, our revised manuscript contains far greater detail on the connection between future AR snowfall, rainfall, and the surface mass balance of the Antarctic ice sheet.

While we have summarized our revisions regarding these topics here, we would point to our in-depth responses to the reviewer comments for more details on all of the above topics. Please find below a response to each reviewer comment in **bold**, with new and revised text in blue.

Sincerely,
Michelle Maclennan and co-authors

Reviewer #1

Reviewer Report

Motivated by the key role the Antarctic Ice Sheet plays in the modulation of present and future global mean sea levels and by the role that precipitation events (in particular synoptic-scale extreme events) play in compensating mass loss from Antarctica to the ocean each year, the authors set out to investigate future changes in Antarctic atmospheric river events and characterize the role they play in the future mass balance of the Antarctic Ice Sheet.

Through the use of the CESM2 large ensemble under the ssp3-7.0 scenario, the authors show that there is projected to be increase in the integrated water vapor (IWV) over the Antarctic continent that represents the largest relative change in IWV at Southern Hemisphere high latitudes. The authors further show that, in addition to the increase in IWV, there is an increase

in poleward integrated vapor transport (vIVT) at Southern Hemisphere high latitudes. Through the use of a polar-specific atmospheric river detection tool, the authors show that there is projected to be an increase in the atmospheric river days per year as a result of the projected changes in IWV and vIVT. To separate thermodynamic and dynamic effects, the authors scale the atmospheric river detection threshold based on the increased IWV, and they find that shifting climatological regions preferential to cyclogenesis lead to smaller increases and decreases in atmospheric river frequency similar to a zonal wave three pattern. Further, the authors note that using this scaling for future atmospheric river detection plays a large role in quantifying the relative role that extreme events like atmospheric rivers play in the overall mass balance and Antarctic contribution to global mean sea level rise.

I find the scope of the work is appropriate for the journal and the analysis of projected changes in Antarctic atmospheric rivers as a result of thermodynamic and dynamic changes to be a worthwhile addition to previous work that has also suggested that future changes in atmospheric rivers are highly sensitive to their detection methods. I do have a few comments that may help clarify some aspects of the manuscript, but as I do not expect my comments to significantly impact the main conclusions of the paper, I recommend minor revisions.

We would like to thank the reviewer for taking the time to review this manuscript and for providing constructive comments. Our responses to the minor comments are below.

Minor Comments

1. I appreciate that the authors include MERRA-2 as a baseline to understand the bias of the CESM2 ensemble, but it piqued my curiosity about whether the changes the authors project in the future climate are visible in observation-based data as well (ie, can you see the increase in IWV as in Fig. 1c but for MERRA-2 data, or is there a trend in AR days per year in MERRA2 based on the current threshold of vIVT for AR detection as in Fig 2c), or is interannual variability too large to get these trends without the use of the ensemble?

We looked into it and find that over the present-day period, there are some trends in AR frequency over Antarctica and the Southern Ocean, but that long-term trends vary heavily by region and the standard deviation in AR frequency over coastal Antarctica is very high (with nearly the same order of magnitude as the annual mean frequency). Below we include a figure showing the annual mean frequency of ARs in MERRA-2, the standard deviation in the annual mean, and the long-term linear trend (pearson linear regression) with black hatching indicating regions with a statistically significant trend ($p < 0.05$). We show results from 1980-2014 since our study uses this period for MERRA-2 to match the CESM2 simulations, but also the 1980-2024 period because this allows us to extend the analysis over the full satellite era, adding 10 years. The 1980-2024 frequency trend (bottom right) shows more regions of statistically significant positive trends in AR frequency than the 1980-2014 period (top right), suggesting that we may be starting to see some of the widespread increases in AR frequency projected in the CESM2 simulations. That said, in the manuscript, we find ubiquitous, large increases in frequency by 2066-2100, whereas over the present-day period there are still certain areas

of frequency decrease (such as over the Amery-Wilkes region of East Antarctica). While the reviewer comment and figure below raise interesting questions about the emergence of trends in AR frequency, the regions of a statistically significant trend in frequency over the Antarctic ice sheet are very small in the present-day, and highly subject to interannual variability of AR events. For example, it is quite likely that events such as the March 2022 AR-heatwave in East Antarctica (Wille et al. 2024) are responsible for damping the strong negative trend in AR frequency over East Antarctica when considering the 1980-2014 trend versus the 1980-2024 trend. Interannual variability is also the hallmark of the present-day impacts of ARs on Antarctic precipitation in MERRA-2, with a 25% ratio between the 1980-2020 AR precipitation standard deviation and mean (Maclennan et al. 2022). The changes we find in AR frequency and precipitation between 2066-2100 and 1980-2014 far exceed the observed trends over the 1980-2024 period, and while this topic would be interesting to examine more closely, it's slightly outside of the scope of this study, and we do not believe it warrants inclusion in the manuscript.

2. Section 2.2: Would it be worthwhile to scale each ensemble member based on its own change in IWV (the individual blue lines in Fig. 1c) opposed to the curves based on the mean, 10th, and 90th percentiles, or is this too noisy due to interannual variability?

We acknowledge that scaling could be applied to individual ensemble members. However, as the reviewer pointed out, the interannual variability in each member is too large and noisy for this approach to be effective. Our goal is to account for the increase in background moisture, which is expected to be a gradual process. Without smoothing, the large interannual variability could introduce moisture changes driven by dynamics rather than background trends. To address this, instead of using a running mean, we

calculate the ensemble mean, which effectively reduces interannual variability while preserving the long-term signal.

3. Section 2.4: I found this section to be quite interesting! However, I was thrown off at first by the description of the projected jet shifts in the Southern Hemisphere as a result of climate change to be 'eastward,' as previous literature (e.g. Kushner et al. 2001; Barnes and Polvani 2013, among others) typically describe the jet response to climate change to be poleward when taken in the zonal mean. After consulting the supplemental material (in particular S13b2), I understand what is meant by an eastward shift in the manuscript as in addition to the well-known polar shift (blue shading poleward of the jet), it appears that the location of the jet max also shifts eastward based on the contours. Perhaps in order to avoid any confusion, it may be preferable to denote this shift as a shift in the maximum jet winds, or a shift in the jet streak, climatologically.

Thank you! As the reviewer mentions, the reason why we describe the projected shift in the polar jet as "eastward" is tied to Supplementary Figure 13, where the region of greatest increase in 500 hPa zonal wind speeds is positioned downstream of the region of maximum zonal wind in the present:

We take the point that being more specific in our wording will help to clarify the result. We have changed the following lines in the manuscript:

L34: "resulting from an eastward shift in the polar jet" → "resulting from an eastward shift in the polar jet maximum wind speeds"

L242: "eastward shift of the polar jet stream" → "eastward shift of the maximum wind speeds in the polar jet stream"

L312: "producing an eastward-shifted jet in the present-day" → "producing an eastward bias in the maximum zonal wind speeds of the polar jet in the present-day"

4. Lines 276-281: While areas of increased poleward 850 flow are co-located with a reduction in sea ice offshore, it appears to me from S15 that reductions in sea ice offshore are present at pretty much all longitudes regardless of changes in poleward flow. Is there relatively more sea ice loss in the regions identified in the manuscript?

Though we find year-round, circumpolar reductions in sea ice in S15, this figure shows that there are certain regions of the Southern Ocean that experience greater sea ice loss than others. In section 2.4, we identify the largest, most intense areas of sea ice loss as the ocean adjacent to the Antarctic Peninsula and Dronning Maud Land in austral summer, the Antarctic Peninsula in austral fall, and Dronning Maud Land in austral winter and spring, often co-located with regions of AR frequency increase in future climates.

In the discussion section (lines 317-327-281), we highlight West Antarctic sea ice loss specifically in reference to a recent study by Naughten et al. (2023). This study suggests that onshore flow, precipitation, and reduced sea ice may be related to increased ocean warming in the Amundsen Sea Embayment in West Antarctica, which has been linked to the retreat and acceleration of glaciers, leading to sea level rise. While this region does not exhibit the largest reduction of sea ice in CESM2 at the end of the 21st century compared to Dronning Maud Land or the Antarctic Peninsula, for example, (S15) we find our results on increased poleward flow, increased precipitation, and reduced sea ice are highly relevant with respect to the Naughten et al. (2023) results, and motivate further study of atmosphere-sea ice-ocean interaction in future climates in this region given its importance for global mean sea level rise.

Supplementary figure 15: CESM2 ensemble mean change in sea ice fraction between 2066-2100 and 1980-2014.

5. Section 4.5: This will likely have no qualitative impact on the manuscript, but as you use a sample of 40 CESM2 LE members (and, truth be told, the 100 total members can be thought of as a sample of all possible realizations, rather than the full population of CESM2 realizations) a

t-statistic may be more appropriate to use in place of a z-statistic, which has a slightly greater critical value for a two-tailed 95% confidence interval with 39 degrees of freedom.

We originally decided on the two-tailed z-statistic given the large sample size of 40 ensemble members. While the full CESM2 large ensemble consists of 100 members, we only consider those with smooth biomass burning (SMBB), meaning from the perspective of this comment, it is more so the case that we're using 40 out of the 50 realizations. We do not consider the 50 ensemble members with CMIP6 forcing (non-SMBB) to be part of the full population because these use biomass burning emissions based on remote sensing data from 1997-2014, which produces large interannual variability in simulations. The choice to use 40 out of 50 SMBB ensemble members came down to the available members with the full set of atmospheric variables used in this study at the time of analysis. We appreciate the reviewer's point that if we were using far fewer samples than the full population (e.g. 40 out of 100), we could not necessarily assume the population variance, meaning the t-test would be more appropriate. However, given that the full set of realizations includes only 10 more ensemble members than we are using, the choice of z-statistic vs t-test has negligible impacts on our results pinpointing regions of statistically significant change. Prior to the submission of this manuscript, we ran a few different statistical tests on our circulation change results to assess their robustness - here are some examples:

We can look at the impact of using a z-statistic vs a t-test for seasonal AR frequency changes in future climates (The red square shows an example region where there are very small differences in stippling, depending on the statistical test and critical value. By and large, there are very few regions where there are differences among the tests):

Future change in AR frequency

Here is a second example, this time for 500 hPa zonal wind:

Future change in 500hPa zonal wind

Z-statistic, alpha = 0.05

Z-statistic, alpha = 0.01

T-test, alpha = 0.05

There is virtually no difference in the regions of change marked as statistically different between the z-statistic and t-test, and reducing alpha from 0.05 to 0.01 within the z-statistic produces negligible differences. These examples suggest the spatial changes in AR frequency and atmospheric circulation are robust, and not easily influenced by the choice of statistical test, nor the threshold.

Other general comments

1. While the manuscript is well-written and clear, there are a couple instances (and I may have missed others) where it looks like words were accidentally omitted or retained in the writing process. Examples of this I noticed were on Line 51, 78-79, and the caption of Fig. 2.

Thank you for flagging these - we have fixed the typos:

L51: We have revised the text to read "there are few studies that examine how these precipitation **extremes may** be impacted..."

L78-79: We have revised the text to read "by the relative increase **in** integrated water vapor..."

Fig. 2 caption: We have revised the text to read "when the present day threshold for AR detection is scaled by **the ensemble mean** relative increase in IWV..."

2. While not necessary, the authors may consider editing the labels in the supplemental material as in some figures there are multiple sets of panels labeled a,b,c, and d, some of which get mentioned in the main text of the manuscript.

Thank you for the recommendation, we have changed the supplementary figure labels to go from (a) to (h).

References

1. Barnes, E. A. & Polvani, L. Response of the Midlatitude Jets, and of Their Variability, to Increased Greenhouse Gases in the CMIP5 Models. *Journal of Climate* 26, 7117–7135 (2013).
2. Kushner, P. J., Held, I. M. & Delworth, T. L. Southern Hemisphere Atmospheric Circulation Response to Global Warming. *J. Climate* 14, 2238–2249 (2001).

Reviewer #2

Reviewer Report

This manuscript examines how atmospheric rivers will affect surface mass balance in Antarctica under future climate change scenarios (SSP3-7.0). Using a set of 40 Community Earth System Model 2 (CESM2) simulations, the authors project that the frequency of ARs could double and their contribution to Antarctic precipitation could triple by the end of the 21st century, using current detection thresholds.

The authors claim that their results show a high sensitivity of projections to AR detection methods, highlighting the need for moisture-based scales for future simulations. The study identifies increased precipitation driven by ARs in coastal regions, particularly in West Antarctica, and highlights the potential interaction of these events with cryospheric and oceanic systems, such as retreating anchor lines in glaciers.

Overall, the paper is very well written and provides interesting results, especially considering the importance of the study region and the scarcity of similar research. However, I am not sure that it reaches the right level for such a high impact journal, and there are also some issues that concern me that I would like the authors to discuss before giving my recommendation for publication.

We thank the reviewer for taking the time to review this manuscript. Our responses to the reviewer comments are below.

Major Comments

1. Although the authors use robust ensembles, they limit their study to the results of a single model, the CESM2. This choice cannot be criticized and is as good a choice as any other. In any case, CESM2 has a number of weaknesses (e.g. high climate sensitivity) that I think should at least be discussed in the manuscript. Of all these, in this case (because of the "similarity" between the two regions) I think the "poor" ability it has shown to resolve the Arctic summer ice cover (see <https://doi.org/10.1029/2020jc016133>) is particularly relevant. I am not too concerned in this case, but I would like to know if the authors have taken this into account; and perhaps include a small discussion of it in the manuscript.

This study focuses on Antarctic AR detection in the CESM2 large ensemble as it enables us to assess how the ensemble spread impacts uncertainty in future AR frequency and precipitation impacts. In the Discussion section (lines 301-317), we address how

circulation and precipitation biases in CESM2 may influence our results on future ARs. In the Datasets section (lines 343-349) we explain our choice to use only ensemble members with smoothed biomass burning (SMBB), as the standard CMIP6 simulations use biomass burning emissions based on remote sensing data from 1997-2014, which produces large interannual variability in simulations. We acknowledge the present-day biases in the CESM2 large ensemble (see Supplementary Figures 2-4, 6, and 11-14 for the seasonal biases of all atmospheric variables studied) and also recognize that future studies implementing a suite of CMIP6 models for Antarctic AR detection will help to further bound uncertainty in future AR frequencies and precipitation impacts.

While we do not perform a sea ice bias analysis in this study, this topic has been heavily addressed in recent studies. For example, Roach et al. (2020) identified that among CMIP6 historical runs, CESM2 is one of the few models (7 out of 40) in which both February (minimum) and September (maximum) sea ice extent are within the range of satellite observations. Figure 1 from Singh et al. (2020) similarly shows that CESM2 sea ice area is closest to observations during the February minimum.

We refrain from discussing CESM2 biases in the Arctic, particularly sea ice, in this manuscript, as the morphology of the Arctic is very different from the Antarctic. Simmonds (2017) highlights these differences: while sea ice in the Arctic Ocean centers over the North Pole and is surrounded by continents, sea ice surrounds the Antarctic continent and extends as far as 55 deg S, well into the westerly storm track. Furthermore, the elevation profile of the Antarctic (rising to 4km in the interior) produces strong katabatic winds at the coast, influencing the size and location of polynyas. There are also different modes of atmospheric variability that affect these two regions, and there are large differences in coastal precipitation and ocean salinity between the Antarctic and Arctic, which all influence sea ice extent. When it comes to model representation of sea ice, DuVivier et al. (2020) finds a key difference between Arctic and Antarctic sea ice representation is that the aerosols necessary for Arctic liquid cloud formation are produced from different precursor emissions and transported to the Arctic - meaning that the transport of cloud-impacting aerosols into the region is much larger in the Arctic than the Antarctic. Ultimately, these strong regional differences suggest that CESM2 sea ice biases in the Arctic should be considered separately from sea ice biases in the Antarctic.

However, we recognize the importance of discussing biases in Antarctic sea ice representation in CESM2, as this also affects precipitation biases over the ice sheet, and have modified the Discussion section (line 305) as follows:

While the CESM2 large ensemble provides a range of uncertainty in future AR projections, we acknowledge that CESM2 model biases may enhance localized patterns of future climate change. The present-day overestimation of Antarctic precipitation by CESM2 may exaggerate future increases in AR-driven snowfall and rainfall (Palerme et al. 2017, Dunmire et al. 2022). The positive bias in precipitation is driven at least in part by the underestimation of sea ice area in CESM2 in the present-day, which promotes

enhanced evaporation from the ocean surface and a resulting overestimation of snowfall over the ice sheet (Trusel et al. 2023). Nevertheless, multi-model analyses suggest robust increases in Antarctic coastal precipitation (up to a 30% increase in total precipitation, and a doubling or tripling of rainfall) at the end of the 21st century in high emissions scenarios, consistent with the findings presented here (Palerme et al. 2017, Vignon et al. 2021).

2. I understand the limitations of the data sources, but I need a discussion on this: is $1.25^\circ \times 0.95^\circ$ an adequate resolution for AR detection? Experience tells me that this is a dangerously low resolution, so I ask the authors to discuss whether its use is justified. It would also not hurt to add a few sentences discussing the pros and cons of ARDT. In particular, I was surprised that the detection method focuses only on the meridional flux of water vapor. Perhaps this is justified in the region of interest, but it is not the norm.

Regarding the grid spacing of CESM2:

There is a host of global studies that rely on the 1-degree grid spacing. Finer spatial resolution would certainly provide higher fidelity representations of AR structure and footprint, which would likely improve our estimates of AR-attributed precipitation as well. That said, the 1-degree model is widely used for AR research and is able to accurately capture large-scale features. This resolution has been applied across past, present, and future climates, globally and regionally, including the Arctic. (e.g., Benedict et al., 2021, O'Brien et al., 2022, Lora et al., 2023, Wang et al., 2024, Rush et al., 2025). We have added the following text to the Discussion to address this (line 295):

Finally, while 1-degree model grid spacing is widely used for AR detection in future climates and is capable of accurately capturing large-scale features (O'Brien et al. 2022, Wang et al. 2024), a finer spatial resolution would provide higher-fidelity representations of AR structure, interactions with topography, and surface impacts (Gilbert et al. 2025), motivating the application of regional climate modeling for future AR studies.

Regarding the choice of ARDT:

In this study we use the Antarctic-specific AR detection tool produced by Wille et al. (2021), which implements the 98th percentile of poleward meridional integrated water vapor transport (vIVT) as the threshold for AR detection. This method is well-documented and heavily used for AR detection in Antarctica, as IVT-based AR detection tools (even those with lower thresholds for the polar regions) fail to capture ARs that penetrate into the interior of the ice sheet, due to the strong zonal winds around the continent (Shields et al. 2022). Therefore, to capture AR precipitation impacts on the ice sheet interior, it is essential to use this AR detection tool. Future development of additional Antarctic AR detection methods that capture ARs over the continent would certainly be helpful in providing estimates of uncertainty due to the choice of AR detection tool.

Figure 1 of Shields et al. (2022), showing the composite difference heatmaps of AR frequency in the Wille catalog relative to (a) global AR detection tools in the Atmospheric River Tracking Method Intercomparison Project (ARTMIP) and (b) AR detection tools in ARTMIP with polar thresholds.

We have added this justification to section 4.2, AR detection, describing the necessity of using this algorithm, on line 387:

This method is well-documented and heavily used for AR detection in Antarctica, as IVT-based AR detection tools (even those with lower thresholds for the polar regions) fail to capture ARs that penetrate into the interior of the ice sheet, due to the strong zonal winds around the continent (Shields et al., 2022).

Minor Comments

Introduction: I fully understand the space constraints of this journal, and the introduction is well done (and coherent). However, I think some relevant bibliography on the role played by ARs in Antarctica is missing and could be cited. For example, take a look at:

<https://doi.org/10.1002/wcc.588>

<https://doi.org/10.3390/w11010041>

Here we would point to lines 62-66 in the Introduction, where we discuss the role of ARs in Antarctic precipitation. To further contextualize the role of ARs in the Antarctic climate system, we have added a sentence:

Despite occurring only 1% of the time at any given location along the Antarctic coastline, these narrow, elongated filaments of extreme water vapor transport are associated with 13% of the annual total precipitation and 35% of interannual variability in precipitation over Antarctica (Wille et al. 2021, Maclennan et al. 2022). Although snowfall is the dominant impact of ARs on the Antarctic ice sheet in the present-day, ARs are also associated with the majority of extreme surface melt events in West Antarctica and on the Antarctic Peninsula, contributing to the destabilization of buttressing ice shelves (Wille et al. 2025).

We have decided not to cite the suggested articles given their focus is on the Arctic and not Antarctica.

L60: I would not say that ARs are the most intense mechanism of moisture transport in the atmosphere. In fact, they are not; they are the major advective mechanism of meridional water vapor transport between the extratropics and mid and high latitudes, which is not the same thing. Also, the commonly used acronym is "ARs", not "ARS".

We have revised the sentence to read:

Here, we focus on the most intense mechanisms for poleward moisture transport in the atmosphere, known as atmospheric rivers.

We have fixed the typo ("ARS" → "ARs").

L74: I congratulate the authors for choosing the SSP370 scenario, which is certainly much more likely and reasonable than SSP585, which for some reason we tend to use heavily in the community. In any case, please include a brief justification for its use in the manuscript.

We chose the SSP3-7.0 scenario as this is the radiative forcing scenario for which the CESM2 large ensemble was run and because it represents a middle-to-high end emissions scenario that is widely regarded as having realistic radiative forcing at the end of the 21st century (Shiogama et al. 2023). Through the use of the CESM2 large ensemble, we had access to 6-hourly output at 32 vertical levels, including all of the variables we required for this study (vIVT, IWV, 2m temperature, sea ice fraction, total precipitation, surface skin temperature, 500 hPa geopotential height, mean sea level pressure, 500 hPa zonal wind, and 850 hPa meridional wind). We have added this justification to line 341 in the Methods:

... (with CMIP6 historical forcing) to future projections from 2015 to 2100 under Shared Socioeconomic Pathway 3-7.0 (SSP3-7.0) forcing (~~mid-high range warming by 2100~~). The SSP3-7.0 scenario is broadly used in CMIP6 model intercomparisons and represents an important pathway of mid-high range warming by 2100 that builds on RCP8.5 (O'Neill et al. 2016, Shiogama et al. 2023).

Fig. 3 Wouldn't it be more logical to put the figure upright instead of on its left side? It is not comfortable for the reader to read vertical text...

We have the figure oriented with present-day precipitation impacts in the top row and the future change in impacts in the second and third row because it matches the structure of fig. 2, which shows present-day AR frequency and frequency bias in the top row and future AR frequency change in the second row. We decided to keep this figure as-is since the vertical text labeling each row is minimal.

Fig. 3, caption. Last line, p-value < 0.1 ??

We have fixed the typo.

L253: Sorry, but how exactly do your results show that? Have different detection methods been applied?

We have changed the word "method" to "threshold" to improve the clarity of this sentence:

Our results show that the future role of ARs in the Antarctic climate system is extremely vulnerable to the detection ~~method~~ **threshold** chosen, which has serious implications for quantifying the influence of extreme events on Antarctic contributions to global mean sea level rise.

Recent studies have shown that the detection threshold - which is the primary difference among different AR detection methods - is the main contributor to uncertainty in future AR frequencies (Shields et al. 2023). This means that uncertainty in the AR detection tool exceeds uncertainty across climate models. In this study, we implement four different detection thresholds for future ARs (the present-day threshold, and scaling the present-day threshold by the 10th percentile, mean, and 90th percentile of the relative increase in atmospheric moisture). By showing that the future role played by ARs in Antarctic precipitation is entirely dependent on the threshold chosen (ranging from 13% to 24% of the total precipitation), we demonstrate how the method of detecting ARs strongly influences the results.

Reviewer #3

Reviewer Report

This article addresses how atmospheric rivers (ARs) may change in Antarctica under future climate conditions, using an ensemble of CESM2 simulations under the SSP3-7.0 scenario. The findings highlight two possible outcomes: if ARs are detected using present-day thresholds, their frequency increases significantly in the future. Conversely, when thresholds are scaled to account for rising atmospheric water vapor due to warming, AR frequency shows only moderate increases or even slight regional declines due to changes in atmospheric circulation. While this result may seem inconclusive to some, the study offers a critical contribution to understanding ARs and their role in Antarctic surface and total ice sheet mass balance. The study projects substantial increases in both snowfall and rainfall over Antarctica as the climate warms. However, attributing these changes directly to ARs depends on the definition of what constitutes an AR.

Overall, I found the manuscript to be methodologically sound, well-written, and supported by clear figures in the main text and supplementary information. The authors provide a careful, nuanced discussion of how ARs and their impacts may evolve, comparing CESM2 with the MERRA-2 reanalysis for present-day conditions and using ensemble-based simulations to explore uncertainty in future projections. This paper addresses an important topic with relevance

to considering the future evolution of the Antarctic ice sheet and its contribution to sea level. I look forward to seeing this article published and I expect that many others in the Antarctic and climate science community would feel the same.

While the paper is strong, there are some areas where the authors could provide additional context and emphasis to enhance the paper's clarity and impact. I thank the authors in advance for considering my comments.

Sincerely,

Luke Trusel

We would like to thank Luke Trusel for taking the time to review this manuscript and for providing constructive comments. Our responses are below.

Broad Comments

I would encourage the authors to consider whether they may be underselling their analysis. The authors might consider emphasizing that extreme events considered rare today will become far more common under future warming. This framing could help underscore the importance of ARs on Antarctica's surface mass balance (SMB). Similarly, greater focus could be placed on the projected increases in both snowfall and rainfall. While rainfall is a smaller component of total precipitation, its relevance for ice shelf stability and the liquid water budget could be more explicitly discussed.

In response to this recommendation, we have modified the language describing precipitation impacts to better emphasize the strong response of AR snowfall and rainfall to changes in climate as follows:

We have strengthened the language around increases in total precipitation starting on line 177 in the Results where, in line with the reviewer comment, we emphasize how future AR-SMB impacts are directly tied to the detection threshold:

This creates a large discrepancy with projected precipitation impacts when only the present-day threshold for AR detection is applied to the 21st century, which results in a **tremendous** ~~substantial~~ increases in AR precipitation across the ice sheet, and particularly in coastal regions and ice shelves. This disparity, driven by the sensitivity of AR detection to increases in atmospheric moisture, has an enormous impact on how we describe the relative importance of ARs in the Antarctic climate system and their future contributions to the surface mass balance of the ice sheet.

On line 215 we added more information on the consequences of increasing AR-attributed rainfall:

Despite rainfall being a small component of the total precipitation in Antarctica, both in present and future climates up to the year 2100, these results highlight the increasingly important role of extreme, synoptic systems in producing rainfall over coastal Antarctica and its ice shelves under the SSP3-7.0 warming scenario. Given the compounding impacts of rainfall in Antarctica, including the raising of firn temperatures through refreezing of liquid water, reduction of surface albedo, firn air depletion, and ice erosion - all of which precondition the surface for widespread melting (Amory et al. 2024) - the increase in AR-associated rainfall may have a considerable impact on future ice shelf stability.

In response to a later comment, we added text to the Discussion section to discuss the advantages and disadvantages of scaling the AR detection threshold in future climates. As part of this, we also strengthen the language around the strong future increase in AR snowfall and rainfall when the present-day detection threshold is used (line 281):

Raising the threshold for AR detection may better preserve the structural consistency (filamentous shape) of ARs in future climates (Payne et al. 2020). However, using the scaled threshold for AR detection to conclude that future ARs retain a comparable role in the Antarctic climate system compared to the present-day undermines the dramatic future increases in AR snowfall and rainfall observed with the present-day threshold (caused by events that would be considered ARs today, but do not meet the future scaled threshold).

Regarding rainfall, specifically, we would like to note that part of our restraint in framing increases in AR rainfall as a major result (in addition to it being a much smaller component of the total precipitation than snowfall) are in relation to the strong positive bias in CESM2 rainfall, which likely has a substantial impact on our results. In response to another comment, we have added text on this as well (line 188):

However, CESM2 exhibits a large positive bias in the total rainfall (up to 175 mm w.e. yr⁻¹) and AR rainfall (up to 25 mm w.e. yr⁻¹) on Bellingshausen and Antarctic Peninsula ice shelves-(Fig. S6). These present-day biases may inflate future projections of liquid precipitation over the Antarctic Peninsula in CESM2 under SSP3-7.0, and therefore estimates of future ice shelf surface mass balance and firn air content reliant only on output from the CESM2 large ensemble may overestimate damage to the snowpack by rainfall.

We feel there is a strong narrative of AR rarity juxtaposed with future increases in frequency throughout the manuscript. For example, in the Introduction (line 63), we write:

Despite occurring only 1% of the time at any given location along the Antarctic coastline, these narrow, elongated filaments of extreme water vapor transport are associated with 13% of the annual total precipitation and 35% of interannual variability in precipitation over Antarctica (Wille et al. 2021, Maclennan et al. 2022).

Then in the Results on (line 126):

... AR frequencies double by the end of the century, with the ice sheet-integrated AR frequency increasing from 15.0 +/- 0.3% (1980-2014) to 31.7 +/-0.5% (2066-2100). AR frequencies increase up to 6 days per year over the Southern Ocean and West Antarctica and 8 days per year over Dronning Maud Land (Fig. 2c). These large, widespread increases in AR frequency suggest a substantial response to future changes in climate compared to the present-day.

And in the Discussion (line 273):

Under the present-day threshold, we find a doubling of AR frequency and a 2.5x increase in precipitation by the end of the 21st century.

By using this type of consistent language in quantifying increases in AR frequency, we aim to underscore the large magnitude of the changes identified.

The focus of the paper is on the SMB response (i.e., snowfall), yet these ARs will also be driving large increases in surface melt (and the liquid water budget overall via rainfall). While the authors do report changes in rainfall, there's no discussion of how melt may change in the future. I understand CESM2 may not represent surface melt directly, but the authors could assess temperature metrics as a proxy for melt. I think equally important to whether ARs will increase in the future is the question of how their impacts will change: will they be mainly positive by delivering enhanced snowfall, or will they drive substantial increases in melt and runoff that may lead to far reduced SMB? I would encourage the authors to give this some thought and consider how explicit analysis of melt or discussion of it might be incorporated in the manuscript.

The reviewer makes an important point and we recognize the value of future AR-driven SMB estimates over the Antarctic ice sheet, as this would enable us to directly quantify the mass balance impacts of future ARs. That said, we would like to explain that the main purpose of this paper is not to quantify the complete SMB impacts of ARs: instead, we are focused on (1) how to detect Antarctic ARs in future climates, (2) how the threshold for detection impacts precipitation contributions, and (3) how circulation changes generate regional shifts in AR frequency, secondary to the strong thermodynamic effect.

Recent studies suggest that the 1-degree grid spacing of CESM2 is too low to capture spatial patterns in surface melt (Noël et al. 2023). While studies such as Dunmire et al. (2022) have quantified overall trends in future SMB in CESM2, the synoptic nature of AR events inherently means that AR-driven surface melting is highly localized and occurs on hourly timescales. A recent regional modeling study by Gilbert et al. (2025) shows how products with 0.25-degree (ERA5 reanalysis) or coarser grid spacing fail to capture the spatial variability of elevated temperatures during AR events, meaning global climate models cannot be relied upon to assess AR-surface temperature relationships in Antarctica. It is for these reasons that we do not consider the CESM2 large ensemble an effective tool to capture the specific surface melt impacts of ARs, nor runoff. Instead, we view this manuscript as the necessary precursor to future, Antarctic-wide studies of

AR-driven changes in SMB using regional climate models and/or statistically downscaled products, as our results provide key information on future AR detection and the implications for precipitation attribution.

To provide some information on AR-associated surface melting in present and future climates, we perform a rudimentary analysis of the frequency of melt conditions associated with AR events. Here we present the annual frequency of surface air temperatures exceeding 0 deg C during AR events (within the footprint of ARs):

Figure S16: The frequency of ARs associated with surface temperatures greater than or equal to 0 deg C in the present-day (1980-2014) and future climate (2066-2100) with the present-day threshold and the mean scaled threshold.

We have added this figure to the supplementary material. This analysis suggests that the Antarctic Peninsula experiences the greatest increase in AR-associated surface melting conditions at the end of the century compared to the present-day, though coastal West Antarctica and Dronning Maud Land also show a frequency increase. However, as we discuss in the manuscript and throughout our responses to reviewers, the vIVT-based ARDT is less effective at capturing AP ARs due to the strong zonal component of moisture transport in this region (Shields et al. 2022). This motivates the use of not only a regional climate model for AR impacts attribution in future climates, but also different detection methods to properly capture the occurrence of ARs over the Antarctic Peninsula, which will affect surface melt and runoff attribution. The future scenario with the present-day AR threshold shows a larger increase in the frequency of AR-melt conditions than the moisture-scaled version, suggesting that the detection threshold for future ARs will have a strong impact on melt attribution. This result goes hand-in-hand with our analysis of AR rainfall with respect to AR-driven surface melt, because if the precipitation is falling as rain, the surface temperature is likely already above or near 0 deg C. Furthermore, any refreezing once rainfall impacts the ice will release latent heat that further warms the immediate environment, increasing the temperature and the potential for surface melt.

It is worth noting that while we do observe substantial increases in the amount of AR-driven rainfall and the frequency of AR-associated surface melt conditions, the increase in snowfall over the grounded ice sheet (which can be inferred from the difference between total precipitation in fig. 3 and rainfall in fig. 4) represents by far the

dominant change, consistent with the analysis by Kittel et al. (2021) of future changes in the total SMB. These results suggest that while AR-driven rainfall and surface melting are likely to increase in future climates, particularly over ice shelves, the primary impact of 21st century anthropogenic warming on Antarctic ARs will be to contribute positively to the SMB of the grounded Antarctic ice sheet. Future studies examining AR changes beyond 2100 or in higher emissions scenarios may observe a future tipping point in which the liquid-to-solid ratio of AR surface impacts increases over the grounded ice sheet, which, if it enhances runoff, may produce a change in the sign of AR impacts on Antarctic mass balance.

Specific Comments

Line 99: Figure 1c effectively highlights uncertainty in future moisture changes. However, other figures (e.g., stippling in Figure 1a) suggest lower ensemble spread, as almost all areas are stippled. A higher threshold for statistical significance (e.g., 2 standard deviations) might better represent regions of true uncertainty.

Yes, we specifically include Fig. 1c to highlight ensemble spread in the change in atmospheric moisture in future climates over time, which cannot be visualized in 1a or 1b. The stippling in 1a and 1b indicates regions where the mean change in IWV (future - present) exceeds the standard deviation in the change in IWV among ensemble members. For clarity, we are not comparing the mean change in IWV to the present-day standard deviation in IWV among ensemble members - in which case testing to determine if the future mean IWV exceeds a higher threshold (such as two standard deviations) would form a more stringent test.

We have tested the change in IWV in multiple ways - for example, supplementary fig. 2e-h shows the seasonal change in IWV with regions of statistically significant change determined using a z-statistic at the 95% confidence interval (testing the difference of means). We observe the same result - a statistically significant increase in IWV across the entire domain - with multiple tests, even with the large ensemble spread. It is even the case when raising the confidence interval from 95% to 99% in the z-statistic (which is quite a high threshold for statistical significance, see results below). This lends confidence to the conclusion that there is high certainty in the increase in Antarctic atmospheric moisture at the end of the century compared to the present-day.

Future change in IWV

Z-statistic, alpha = 0.05

Z-statistic, alpha = 0.01

We recognize that the text on line 107, emphasizing the ensemble spread, is slightly juxtaposed with these results. Therefore, we have modified the text as follows, to reflect that the magnitude of the relative change in IWV varies among ensemble members even with the statistically significant increase at the end of the 21st century compared to the present-day:

The 10th and 90th percentiles of IWV increase range from 1.3-2.1x present-day values, demonstrating high-uncertainty in the magnitude of the relative future increase changes in atmospheric moisture even within the single radiative forcing scenario of SSP3-7.0,

Line 114: Consider adding "in these regions" to the end of the sentence for clarity

Done!

Line 117: Please clarify the discrepancy between the 15% AR frequency reported here and the 1% frequency mentioned in the introduction. Figure S5d also suggests ~3% frequency without IWV ratio scaling, which adds to my confusion.

Yes, we're happy to clarify this. When we refer to a 1% frequency of ARs in the introduction, this means that the average annual frequency of ARs at any given location along the coastline is 1% (an individual grid-cell value). We use this metric in the introduction because this is the frequency value first presented in Wille et al. (2021) and consistently referenced in Antarctic AR literature. When we refer to AR frequency increasing from 15% to 32% on line 128, we use the term "ice sheet-integrated AR frequency". This value is calculated by determining the number of timesteps per year there is an AR anywhere over Antarctica (including ice shelves) and dividing by the total number of timesteps per year. This means that 15% of the time there is an AR making landfall somewhere over Antarctica in the present-day. It's a new metric for the Antarctic-wide frequency of ARs, but builds on the method presented in MacLennan et al.

(2023) to quantify the "total" frequency of ARs in the Amundsen Sea Embayment - Marie Byrd Land region. The ice sheet-integrated method is a helpful way of encapsulating total changes to AR frequency over the ice sheet, given that coastal changes in frequency vary strongly by region.

We have revised the sentence in the introduction to be more explicit:

"Despite occurring only 1% of the time at any given location along the Antarctic coastline,"

Now this ^ contrasts better with the language in the results on line 128:

"... with the ice sheet-integrated AR frequency increasing from 15.0 +/- 0.3% (1980-2014) to 31.7 +/- 0.5% (2066-2100)."

To make sure this difference is clear to readers, we added a short subsection to the Methods, "Calculating AR frequency":

We use two metrics to quantify AR frequencies over the Antarctic ice sheet. First, we use the grid cell frequency of ARs to describe their occurrence at any given location over the Southern Ocean or the Antarctic continent. We calculate the grid cell frequency by summing the time steps with AR occurrence at each location in the domain and dividing by the total time steps. Second, we also describe the ice sheet-integrated annual frequency of ARs (including ice shelves). This value is calculated by counting the number of time steps per year there is an AR landfall anywhere over Antarctica (includes ice shelves, and time steps where multiple ARs make landfall at once) and dividing by the total number of time steps per year.

Regarding fig. S5, here we show the future frequency of ARs from 2066-2100 (see beginning of the figure caption), with frequencies from all four future thresholds on the same color scale. So (d) shows the future frequency of ARs when you use only the present-day threshold for detection. It's the pattern you would get if you summed fig. 2a (present frequency) and 2c (future change in AR frequency with the present-day threshold) in the main text.

Figure S5: CESM2 ensemble mean annual AR frequency from 2066 to 2100 based on scaling for the increase in IWW

with (a) the 90th percentile of the IWV ratio (most restrictive), (b) the mean IWV ratio, (c) the 10th percentile of the IWV ratio (less restrictive) and (d) no scaling, only using the present-day threshold for AR detection.

Line 134: Probably good to cite Raphael 2004 with respect to zonal wave three:

Raphael, M. N., 2004: A zonal wave 3 index for the Southern Hemisphere. *Geophys. Res. Lett.*, 31, L23212, <https://doi.org/10.1029/2004GL020365>.

Done!

Figure 2 caption: Correct the text: "scaled by thees relative average relative" to "scaled by these relative average changes in IWV." Also, explicitly reference Figure 1b if relevant.

We have fixed the typo:

(d) Future AR frequency difference in CESM2 (2066 to 2100 minus 1980 to 2014) when the present-day threshold for AR detection is scaled by ~~thees relative average~~ **ensemble mean** relative increase in IWV ~~among all ensemble members~~.

We may refrain from referencing fig. 1b here because, while it shows the ensemble mean change in IWV, here we're describing scaling by the mean trend line in 1c, but for every point in the domain (this is the curve for one location, and they vary slightly by grid cell).

Lines ~156-160: The authors remain neutral on whether AR detection thresholds should be scaled for future IWV increases. Adding recommendations or arguments for and against scaling would be helpful. For instance, if the goal is to classify ARs consistently under changing climates, scaling might be necessary.

The objective of this study is to show how sensitive AR frequency, and in particular the attributed precipitation impacts, are to the choice of AR threshold. We were neutral on whether AR detection thresholds should be raised in future climates or remain the same because it is a highly science-specific decision. From the synoptic perspective, to retain consistency in the structure, footprint, and extreme-ness of ARs in future climates, raising the threshold may be preferable. In this study, we raise the present-day vIVT threshold only in accordance with the increase in atmospheric moisture, which enables us to separate the thermodynamic effect on ARs from changes in atmospheric circulation. Other studies use a percentile-based threshold based on the future climate, meaning that both thermodynamic and dynamic changes impact the chosen threshold (Payne et al. 2020).

Comparatively, future studies on Antarctic surface mass balance and the role of extreme events may be better suited to no scaling (the present-day threshold). By using the moisture-scaled threshold, you may eliminate some events that would be considered ARs today because they don't meet the future threshold. However, that doesn't mean that these intermediate events (ARs today, but not extreme enough to qualify in the future) don't occur and cause precipitation or surface melt in future climates. By stating that the

role of ARs in the Antarctic climate system is comparable to the present-day in future climates under a scaled threshold, we risk undermining consequential increases in the surface impacts of these intermediate events. That is why we emphasize so heavily throughout the manuscript that the future role of ARs in the Antarctic climate system is extremely vulnerable to the detection method chosen - and ultimately, this is the main position we want to take. We have added text to the Discussion to address some of these nuances (line 278) :

Our results show that the future role of ARs in the Antarctic climate system is extremely vulnerable to the detection method chosen, which has serious implications for quantifying the influence of extreme events on Antarctic contributions to global mean sea level rise. Raising the threshold for AR detection may better preserve the structural consistency (filamentous shape) of ARs in future climates (Payne et al. 2020). However, using the scaled threshold for AR detection to conclude that future ARs retain a comparable role in the Antarctic climate system compared to the present-day undermines the dramatic future increases in AR snowfall and rainfall observed with the present-day threshold (caused by events that would be considered ARs today, but do not meet the future scaled threshold). This suggests the choice of future AR threshold is a science-specific decision that depends on the systems considered, just like the choice of AR detection tool (Shields et al. 2022). Our ~~These~~ findings are largely consistent with global studies of future ARs, which suggest a strong response of ARs to thermodynamic forcing in high emissions scenarios resulting in a poleward shift in AR frequency in the Southern Hemisphere (Ma et al. 2020)....

Line 174/195: I am concerned over the large magnitude of rainfall in present-day CESM2, especially on the Antarctic Peninsula (Figure S6). I think more discussion of this bias is warranted. Given that ARs are also relevant to the liquid water budget (especially in the future), how this bias might impact projections of future SMB change or ice shelf instability is warranted. Line 195 somewhat downplays the rainfall bias as being a small component of the total precipitation. Yes, this is true, but rainfall here is a large fraction of the liquid water budget (e.g., melt rates today on the northern AP are ~400 mm w.e./yr and CESM2 suggests ~200 mm w.e./yr rainfall here in the present-day).

We appreciate this feedback from the reviewer and have revised the text on lines 185-197 to strengthen the discussion of present-day rainfall biases in CESM2 and the implications for projections of future rainfall:

In the present-day, ARs contribute up to 25 mm w.e. yr⁻¹ of rainfall in coastal Antarctica (up to 30% of the total, Fig. 4a,b), particularly along the Bellingshausen Sea, and explain 40-75% of rainfall variability across most major ice shelves (Fig. 4c). However, CESM2 exhibits a large positive bias in the total rainfall (up to 175 mm w.e. yr⁻¹) and AR rainfall (up to 25 mm w.e. yr⁻¹) on Bellingshausen and Antarctic Peninsula ice shelves, ~~though the relative contribution of ARs to the total rainfall is lower than MERRA-2 by more than 40% in these regions~~ (Fig. S6). These present-day biases may inflate future projections

of liquid precipitation over the Antarctic Peninsula in CESM2 under SSP3-7.0, and therefore estimates of future ice shelf surface mass balance and firn air content reliant only on output from the CESM2 large ensemble may overestimate damage to the snowpack by rainfall. The relative contribution of ARs to the total rainfall in CESM2 is lower than MERRA-2 by more than 10% over Antarctic Peninsula ice shelves, indicating the role of ARs in contributing rainfall to this region is reduced in CESM2 compared to MERRA-2.

With these edits, we also aim to clarify the final sentence in the above paragraph, which was not intended to downplay rainfall biases. Here, we try to highlight the connection between the large positive bias in total rainfall and AR rainfall and negative bias in the relative contribution of AR rainfall to the total rainfall over the Antarctic Peninsula. These results suggest that the bias in AR rainfall is not the primary contributor to the bias in total rainfall in CESM2 and that the role ARs play in bringing rainfall to this region is smaller in CESM2 than in MERRA-2.

Section 2.4 / Line 212 / Line 223: There's some discussion of sea ice here, but not much context for its relevance. Is the sea ice responding to the changes in atmospheric circulation? Are ARs in the model causing reduced sea ice? Is the reduced sea ice leading to more moisture? Some context would be helpful.

We appreciate the recommendation to include more analysis of the sea ice results presented in this study. First, in response to a comment from another reviewer, we have addressed how low sea ice biases in CESM2 may contribute to positive precipitation biases in CESM2 in lines 301-311 of the Discussion:

While the CESM2 large ensemble provides a range of uncertainty in future AR projections, we acknowledge that CESM2 model biases may enhance localized patterns of future climate change. The present-day overestimation of Antarctic precipitation by CESM2 may exaggerate future increases in AR-driven snowfall and rainfall (Palerme et al. 2017, Dunmire et al. 2022). The positive bias in precipitation is driven at least in part by the underestimation of sea ice area in CESM2 in the present-day, which promotes enhanced evaporation from the ocean surface and a resulting overestimation of snowfall over the ice sheet (Trusel et al. 2023). Nevertheless, multi-model analyses suggest robust increases in Antarctic coastal precipitation (up to a 30% increase in total precipitation, and a doubling or tripling of rainfall) at the end of the 21st century in high emissions scenarios, consistent with the findings presented here (Palerme et al. 2017, Vignon et al. 2021).

While future studies specifically analyzing the present-day and future relationship between Antarctic ARs and sea ice would be optimal for quantifying this relationship, we recognize the importance of this topic and address it on line 321 in the Discussion:

In East Antarctica, the co-location of dynamically-driven increases in AR frequency with strong sea ice decline offshore Dronning Maud Land in austral winter and spring may signal a positive feedback loop where reduced sea ice extent enhances moisture availability for Antarctic ARs, while the increased frequency of ARs and associated downward longwave radiation and rainfall exacerbates sea ice melting (Zhang et al. 2023).

Line 220 (and elsewhere regarding polar jet): Please clarify how changes in geopotential height reflect the eastward shift of the polar jet. Additional explanation of the methods used to determine jet stream changes would improve clarity.

In this section we highlight the future increase in 500 hPa geopotential heights over the Southern Ocean from 90 deg E to 110 deg W, which drives an increase in the thermal wind (Shaw et al. 2024), resulting higher zonal wind speeds directly poleward of the region of the largest 500 hPa geopotential height increase. This is to the east of the present-day maximum zonal wind speed in the polar jet (supplementary figure 13). In response to a similar comment from Reviewer #1, we have modified the text as follows, to clarify how we derive this conclusion:

L33: "resulting from an eastward shift in the polar jet" → "resulting from an eastward shift in the polar jet maximum wind speeds"

L220: "eastward shift of the polar jet stream" → "eastward shift of the maximum wind speeds in the polar jet stream"

L271: "providing an eastward-shifted jet in the present-day" → "providing an eastward bias in the maximum zonal wind speeds of the polar jet in the present-day"

Line 231: Discussion here how there's an increase in zonal winds and this impacts the detection of ARs using the (meridional) vIVT method. It could be mentioned that other ARDTs might produce better classifications (e.g., ones based on IWV or uIVT).

Thank you the recommendation, we have added a sentence on line 251:

AR frequencies decrease over the Antarctic Peninsula, likely resulting from a poleward shift of the storm track and enhanced zonal flow over the Peninsula (Fig. S13g), exacerbating present-day biases in the AR detection tool, which is tuned to capture meridionally-propagating ARs. **Therefore, we note that AR detection methods based on IWV or total IVT (both meridional and zonal water vapor transport) may produce better classifications in this region when considering the effect of circulation changes on future AR frequency (Shields et al. 2022).**

Reviewer #4

Reviewer Report

The manuscript “Rising atmospheric moisture escalates the future impact of atmospheric rivers in the Antarctic climate system” by Michelle Maclennan et al. analyses the sensitivity of Antarctic atmospheric rivers (ARs) to future changes in atmospheric moisture in terms of AR frequency and AR-related precipitation. For this purpose, they compare the present-day climate (1980 – 2014) and the future climate (2066 – 2100) based on the MERRA-2 reanalysis product and the global climate model CESM2 LE. They generally find an increase in AR frequency and precipitation impacts; however, these results are sensitive based on the detection algorithm and the water vapor increase. Finally, they analyze the linkage between circulation changes and atmospheric river activity.

Since the ARs play an important role in the Antarctic Ice Sheet, it is of particular interest to investigate how the AR occurrence and the AR-related precipitation will change in the future. Therefore, this study gives a good overview of future changes and sensitivity in AR-frequency and AR-related precipitation in the Antarctic.

The manuscript is well written, and I think it is a valuable study for the AR community and the analysis of future changes in Antarctica. Further, it is helpful to better assess the changes in the Antarctic ice sheet related to AR precipitation. However, I have some questions about the methodology and suggestions for improving the structure and figures.

The authors would like to thank the reviewer for taking the time to review this manuscript and for providing constructive comments. Our responses to specific comments are provided below.

Specific Comments

In the last sentence of the abstract (LL 33 – 36), you mention the importance of using large ensembles compared to few ensemble members. This statement sounds logical. However, I cannot find analyses that compare large ensemble members to a few.

We highlight the importance of using large ensembles in the abstract in reference to section 2.1, the atmospheric moisture increase. In Figure 1c, we show large variability among the 40 ensemble members with respect to the relative increase in integrated water vapor (I WV) over the Antarctic continent - variability that may not be captured if only one or two members are used for an analysis of future changes in I WV. We have revised the last few sentences of this section, and added more details, to illustrate this point:

An exponential increase in atmospheric moisture occurs over the Southern Ocean and Antarctica over the 21st century relative to the present-day, strengthening the pole-equator moisture gradient (Fig. 1a). The greatest increase in ensemble mean I WV (+5 kg m⁻²) occurs over the Atlantic Ocean east of South America, the Indian Ocean equatorward of the Amery Ice Shelf, and the Pacific Ocean equatorward of the Ross Sea

(see Fig. S1 for map of locations). The ensemble mean IWV increases less than 1 kg m^{-2} over the Antarctic continent, which is characterized by extremely cold and dry air (Fig. S2). However, this region is marked by the greatest relative increase in IWV (1.5 x present-day) at the end of the 21st century (Fig. 1b). ~~Large variability in the IWV increase among the 40 ensemble members (the 10th and 90th percentiles range from 1.3-2.1 x present-day values) indicates high uncertainty for future moisture changes within the SSP 3-7.0 scenario (Fig. 1c).~~

There is large variability ($\sim 2 \text{ kg m}^{-2}$) in the future IWV increase among the 40 ensemble members, with extremely high IWV ratios during individual months from 2015 to 2100, which can reach 3-3.5 times the present-day climatology by the end of the 21st century (Fig. 1c). The 10th and 90th percentiles of IWV increase range from 1.3-2.1 x present-day values, demonstrating high uncertainty in future changes in atmospheric moisture even within the single radiative forcing scenario of SSP3-7.0, and underscoring the advantage of using many ensemble members to capture this range of variability.

Can you better explain the calculation of the IWV ratio in equation (2)? It becomes more clear in the caption of Figure 1c) but here, the IWV ratio varies for each year. Thus, it is still unclear which value you used for the IWVratio future.

Equation (2) shows how we calculate the threshold for atmospheric river (AR) detection in future climates when scaling for the relative increase in atmospheric moisture. Here $AR \text{ threshold}_{future}$ represents the future threshold for ARs, $IWV \text{ ratio}_{future}$ represents the 10th percentile, ensemble mean, or 90th percentile relative increase in IWV, and $vIVT \text{ threshold}_{present}$ represents the present-day threshold of meridional integrated water vapor transport (vIVT) used to detect ARs, based on Wille et al. 2021 (doi: 10.1029/2020JD033788).

$$AR \text{ threshold}_{future} = IWV \text{ ratio}_{future} * vIVT \text{ threshold}_{present} \quad (2)$$

The method for calculating $IWV \text{ ratio}_{future}$, and generating the moisture scaling for the AR detection threshold, is described on lines 411-448. Essentially, $IWV \text{ ratio}_{future}$ represents a scaling curve based on the relative increase in atmospheric moisture at the 10th percentile, ensemble mean, and 90th percentile level. The value of the ratio is determined by the three trendlines in Fig. 1c. The value of the ratio does increase every month, meaning that under the ensemble mean scaling curve, for example, the AR detection threshold in January 2095 will be higher than the threshold in January 2070, in accordance with increasing amounts of atmospheric moisture.

We have revised the text to make it more clear how this ratio is calculated:

Then, for each grid point, we produce a time series of the relative monthly increase in IWV compared to the historical climatology for each ensemble member from 2015 to 2100. We determine the 10th percentile, mean, and 90th percentile of the relative IWV

increase amongst CESM2 the ensemble members for each month, to capture the range of variability in future moisture projections in CESM2. To generate smooth IWV scaling curves, we then take a 10-year running mean of the 10th percentile, ensemble mean, and 90th percentile curves and then calculate a third-degree polynomial trend line for each curve (Fig. 1c). These three curves determine the scaling mechanisms used to modify the AR detection threshold to account for increases in atmospheric moisture in the future climate, and increase over time from 2015-2100 in accordance with atmospheric moisture.

We apply each scaling level - the 10th percentile, ensemble mean, and 90th percentile trend line IWV ratios - to the present day vIVT threshold for AR detection from 2066 to 2100 for each grid point, as follows. Here $AR\ threshold_{future}$ represents the future threshold for ARs, $IWV\ ratio_{future}$ represents the moisture scaling factor (10th percentile, ensemble mean, or 90th percentile relative increase in IWV), and $vIVT\ threshold_{present}$ represents the present-day threshold of vIVT used to detect ARs.

Thus, we develop four thresholds for future Antarctic AR detection using different levels to account for uncertainty in the increase in atmospheric moisture: (1) no scaling (the present day vIVT threshold), (2) the ensemble 10th percentile of the relative increase in IWV among ensemble members (less restrictive than the mean increase), (3) the ensemble mean relative increase in IWV, and (4) the ensemble 90th percentile of the relative IWV increase (more restrictive than the mean increase). We then run the ARDT on CESM2 vIVT from 2066 to 2100 with the updated AR thresholds. For the presentation of results, we primarily focus on the difference between (1) no scaling and (3) scaling by the mean relative increase in IWV, though we provide frequency and precipitation analyses for all scenarios in the supplementary figures. For the frequency analysis presented in the main text, we compare AR frequencies between the future period (2066-2100) and the present day (1980-2014), and mark regions with statistically significant differences wherever the absolute value of the mean difference in AR frequency between periods exceeds the standard deviation among CESM2 ensemble members.

For the future climate, you use the CESM2 LE. Why are you using this model? Is there a specific reason? For the model uncertainty, you compare it with E3SMv2. Would it make more sense to use a multi-model ensemble? How did you choose the 21 ensemble members for CESM for the comparison with E3SMv2?

We use the CESM2 large ensemble because it enables us to examine ARs in the present and future climate (up to 2100) in 40 ensemble members at 6-hourly temporal resolution with 32 vertical levels, including all of the variables we required for this study (vIVT, IWV, 2m temperature, sea ice fraction, total precipitation, surface skin temperature, 500 hPa geopotential height, mean sea level pressure, 500 hPa zonal wind, and 850 hPa meridional wind). Although applying a full CMIP6 multi-model ensemble was beyond the scope of this work, we address model uncertainty by comparing two different models,

with different ocean components, different model biases, and different variability and climate change responses (Fasullo et al., 2024, Meehl et al., 2024). The E3SMv2 large ensemble was only comprised of 21 members thus constraining the number of ensemble members for this comparison.

For the analysis, you use 40 ensemble members from CESM2 LE. Initially, I wondered why you chose these 40 ensemble members. Later, you mention that you use the ensemble members with smoothed biomass burning. I think you should mention this earlier. Can you also explain why you chose these ensemble members?

We use 40 ensemble members from the CESM2 LE with smoothed biomass burning because these between represent emissions in the present-day period compared to standard CMIP6 simulations, have 6-hourly output at 32 vertical levels, and include all of the variables we required for this study (vIvT, IWV, 2m temperature, sea ice fraction, total precipitation, surface skin temperature, 500 hPa geopotential height, mean sea level pressure, 500 hPa zonal wind, and 850 hPa meridional wind). Section 4.1.1 describes the dataset and ensemble members used; we could move the description of our choice of smoothed biomass burning members up by one sentence (switching it with the sentence in green below), but we feel that is it helpful to mention our application of the CESM2 LE as a future vs present-day comparison first, as the smoothed biomass burning refers to the specific nature of CESM2 LE output in the present-day.

The CESM2 Large Ensemble (CESM2 LE) consists of 100 ensemble members spanning 1850 to 2100 (Rodgers et al. 2021, Danabasoglu et al. 2020). CESM2 LE is a fully coupled atmosphere-ocean-land model and uses the Community Atmosphere Model Version 6 (CAM6) with 32 vertical levels and a model top at 2.26 hPa (about 40 km). Here we use 40 ensemble members from CESM2 LE with a high (6-hourly) temporal resolution and 1.25 deg longitude by 0.95 deg latitude horizontal grid spacing. *We compare the present-day period from 1980 to 2014 (with CMIP6 historical forcing) to future projections from 2015 to 2100 under Shared Socioeconomic Pathway 3-7.0 (SSP3-7.0) forcing (mid-high range warming by 2100).* Standard CMIP6 simulations use biomass burning emissions based on remote sensing data from 1997 to 2014, which introduces greater interannual variability in simulations and can impact future projections of the climate response to atmospheric forcing (Derepentigny et al. 2022, Fasullo et al. 2022). Instead, we use ensemble members with smoothed biomass burning (SMBB), which uses a biomass burning protocol from 1997 to 2014 that is more comparable to emissions used prior to 1997.

I am not sure if it would be better to swap chapters 2.3 and 2.4. I think chapter 2.4 is more related to 2.2.

We decided to keep the current order of chapters 2.3 and 2.4, as we feel strongly that the differential AR precipitation impacts depending on the future AR detection threshold

represents the most important result from our study, and it follows well from 2.1, where we introduce the moisture scaled threshold in comparison to the present-day threshold for AR detection. Because dynamical changes in AR frequency are secondary to the thermodynamic effect, and this is reinforced by the differential precipitation impacts presented in 2.3, we feel that it makes sense to present the circulation changes afterwards.

Comments to Figures

Figure 1a): It might be better to use a colormap with more contrast.

We have updated the colorbar on Fig. 1a to exhibit more contrast between areas of strong atmospheric moisture increase (dark blue) and areas of no change (white), while keeping the sequential colormap. Here, we altered the colorbar values from 0 to 5.1 with 0.1 spacing to values from 0 to 4.6 at 0.2 spacing (both are with extension at the maximum). The maximum values of moisture increase are still 5 kg m^{-2} (not changing the dataset plotted) but this new version better emphasizes the areas of large moisture increase.

Figure 1c): It is difficult to see the mean trend line. I would change the color for the ensemble mean or the mean trend line.

We modified the ensemble mean line color to a brighter blue and thickened the black lines for each scaling curve. Additionally, we relabeled the black lines as "scaling curves" (previously "trend") as it was mentioned in an earlier comment that it was unclear what was used for the IWV ratio.

It is a bit confusing that you are not consistent with the colormap. For example, in Fig. 2, negative values are red, and positive values are blue in Figs. S3, S11, and S12, it is reversed.

We strive to use intuitive colormaps for the variables plotted, which does result in some cases where negative values are red and other cases where negative values are positive. For example, when considering precipitation in fig. 3 and 4, we use blue and green colors to indicate higher precipitation, and white to indicate lower precipitation. This means that when we plot precipitation biases with a diverging colormap in S6, it follows that we use blue to represent higher precipitation and red for lower precipitation. Since the dominant impact of ARs in Antarctica is on precipitation, we feel that it is important for science communication to align the colormaps, such that higher AR frequencies are in blue and lower AR frequencies are in red (fig. 2 and S3, and S14 for poleward winds). Similarly, the standard for mean sea level pressure is to use blue for lower pressure and red for higher pressure/higher geopotential heights (S4 and S11). Since the polar jet stream (500 hPa zonal wind speed) is so closely tied to the low pressure system offshore the Ross Sea, we use blue to indicate regions of zonal wind speed increase in S13. It makes sense to use red to indicate regions of temperature increase in S12 and blue for temperature decrease.

Interestingly, you have a significant change for all grid points in Fig. 1a) and b), Fig. 2c), and also in the supplementary.

Yes - and even if we raise the confidence interval (in response to a comment from Reviewer #2), we still find a statistically significant increase everywhere (below). We include the stippling because it's a helpful visual tool to emphasize this.

Future change in IWV

Z-statistic, alpha = 0.05

Z-statistic, alpha = 0.01

Also in response to the earlier comment, we recognized that the text on line 107, emphasizing the ensemble spread, is slightly juxtaposed with these results. Therefore, we have modified the text as follows, to reflect that the magnitude of the relative change in IWV varies among ensemble members even with the statistically significant increase at the end of the 21st century compared to the present-day:

The 10th and 90th percentiles of IWV increase range from 1.3-2.1x present-day values, demonstrating high-uncertainty in the magnitude of the relative future increase changes in atmospheric moisture even within the single radiative forcing scenario of SSP3-7.0,

Supplementary: For most images, you distinguish between left and right and use a-d for each image. I would prefer the letters a-h. I think that would make it a bit clearer.

Other reviewers had similar feedback. We appreciate the recommendation and have amended our supplementary figures to use a-h labeling.

I like Figure 5: It gives a very good overview!

Thank you!

Minor comments

L29: In the discussion (Chapter 3 LL 248-249), the increase in precip is three times higher

Thank you for flagging this typo in the discussion - AR frequency over the Antarctic ice sheet increases from 15% to 32% and AR precipitation increases from $\sim 400 \text{ Gt yr}^{-1}$ to $\sim 1000 \text{ Gt yr}^{-1}$.

We have revised the discussion (line 247) as follows:

Under the present-day threshold, we find a doubling of AR frequency and a ~~near three-fold~~ 2.5x increase in precipitation by the end of the 21st century.

L30: which surface impacts? Only precip? At least you apply the algorithm only to precipitation. The changes in other surface variables (500 hPa geopotential height, 2m temperature, 500 hPa zonal wind...) are not related to the threshold.

Here we used "surface impacts" because the manuscript addresses total precipitation and rainfall, and because precipitation is the dominant impact of ARs on the Antarctic ice sheet surface. We have revised the sentence as follows:

However, future ~~surface~~ precipitation impacts are critically dependent on the detection threshold for ARs: accounting for moisture increases in the threshold produces smaller, regional changes in AR frequency and precipitation, primarily resulting from an eastward shift in the polar jet maximum wind speeds.

L61: ARS → ARs

Corrected it, thanks.

L78: present day → present-day

Corrected it, thanks.

LL94-95: I would relocate Fig. S2 after the Antarctic continent in L94. In this position, I would expect a temperature map in Fig. S2.

We originally had the figure reference at the end of the sentence because the figure shows both the small increase in IWV over the continent in future climates as well as the dry air in the present day. We moved it to after "Antarctic continent":

The ensemble mean IWV increases less than 1 kg m^{-2} over the Antarctic continent (Fig. S2), which is characterized by extremely cold and dry air (~~Fig. S2~~).

L103: Here you write: CESM2 large ensemble. Sometimes, you write CESM2 or CESM2 LE → Is CESM2 LE all ensembles and CESM2 only the 40 ensemble members?

We have revised the text to exclude the usage of "CESM2 LE" to reduce the number of acronyms used. In the introduction, we introduce the dataset as the "CESM2 large ensemble". Hereafter, we refer to the output as "CESM2".

We occasionally use "CESM2 ensemble members" - only when we specifically refer to the spread or uncertainty in results. For example, in the discussion (line 301):

While the CESM2 large ensemble provides a range of uncertainty in future AR projections, we acknowledge that CESM2 model biases may enhance localized patterns of future climate change.

In the methods (Global Climate Models) section, we introduce the product used as the CESM2 large ensemble, and we use the term "large ensemble" in reference to both CESM2 and E3SMv2. Similar to the format of the rest of the manuscript, we drop "large ensemble" hereafter in the methods.

L109: the low-pressure bias is only seen in Fig. S3; Fig. S4 shows the difference between CESM and E3SMv2 → mention Fig. S4 earlier in L108 in addition to Fig. 2b

We switched the order of S3 and S4 in the supplementary material and now include the figure references as follows:

CESM2 exhibits a positive bias in AR frequency of 0.5 days yr⁻¹ offshore of West Antarctica and over the Ross Ice Shelf (Figs. 2b, S3), which may be explained by a low-pressure bias of CESM2 in this region, conducive to poleward moisture transport (Figs. S3, S4).

L112-114: Do you think this blocking system is responsible for an AR increase over the Ross Sea?

Here we show that directly poleward of the low-pressure biased regions in CESM2, there are regions with lower AR frequencies in the present-day compared with MERRA-2. For the area offshore of Wilkes Land and the Ross Sea, we show in section 2.4 that this is a region of strong increase in 500 hPa zonal wind in the future climate, representing an eastward shift in the maximum intensity of the climatological polar jet. CESM2 does exhibit an eastward bias in 500 hPa zonal wind in the present-day, which is likely responsible for producing the low-pressure bias offshore of Wilkes Land, and likely the lower AR frequencies in this region over the continent. An intensification and eastward shift of the polar jet as seen in CESM2 would position the right exit region, an area conducive to upward vertical motion, directly over the Ross Sea - and we suggest in section 2.4 that this may be the mechanism driving increased AR frequencies here in future climates.

L121: present day → present-day

Corrected it, thanks (replaced all instances of "present day" with "present-day" in the manuscript).

L136: (Gt) → Gt

This is the first time we use this unit so we have written it out as "Gigatons (Gt)".

L140: ARs explain 3—50% of the interannual variability → Change colormap in Fig. 3c, maximum 50%

Previously the colorbars for Fig. 3c and 4c ranged from 0 to 100% contribution to year-to-year variability in precipitation and rainfall, respectively. We have adjusted the colorbars of Fig. 3c and 4c to reflect the maximum percent variance explained for each variable, respectively.

L145: Fig. S6 → (Fig. S6)

Corrected it, thanks.

LL147-148: change colormap from -10 to 10%

Thank you for the recommendation, we have changed the range of the colormap on supplementary fig. 6 as suggested:

We modified the text on line 155 to better reflect the regions of largest difference:

However, CESM2 puts less importance on AR-related precipitation than MERRA-2: the bias in the relative contribution of ARs to the total precipitation ranges from -6 7%

(coastal Dronning Maud Land and Wilkes Land) to 10 9% (Antarctic interior Dronning Maud Land).

LL152-153: Increases in AR precipitation outpace increases in total precipitation → cannot see this in Fig. S7. Do you mean Fig. S8

The reference to Fig. S7 in this sentence refers to "increases in total precipitation", which can be seen in the figure. The idea is that readers can see changes in AR attributed precipitation in Fig. 3d (referenced in previous sentence) and compare to the changes in total precipitation in Fig. S7, if they like. The comparison between Figs. 3b and 3e shows the rise in the relative contribution of ARs to the total precipitation, but comparing S7 and 3d provides a more direct comparison.

LL 154: 24 +/- 2% in Fig. 3e → adapt colormap

This is an ice-sheet integrated value; we have revised the text to mention this:

Increases in AR precipitation outpace increases in total precipitation (Fig. S7), such that the relative contribution of ARs to the total rises to 24 +/- 2% (ice sheet-integrated value, see Fig. 3e for regional changes)...

LL160: 'No change'? → There is a change but not so high – maybe more precise: there is no significant change?

We aim to only discuss statistically significant changes in this manuscript, which is why we used the phrase "no change" here. We revised the sentence:

The relative contribution of ARs to the total precipitation remains at 13 +/- 0.3%, and there is no significant change in the contribution of ARs to precipitation variability (Fig. 3h,i)

LL164-167: In which Figure can you see this?

This is a summary of the second part of the previous paragraph (lines 159-174), where we discuss in detail how scaling the AR detection threshold in accordance with the ensemble mean increase in atmospheric moisture yields a similar role of ARs in contributing to Antarctic precipitation at the end of the 21st century as in the present-day (with all of the figure references). We have merged this paragraph with the previous paragraph, so that it's clearly associated with the prior results, and added the word "Thus" to indicate that the sentence acts as a conclusion to this set of results:

"... and an increase in their relative contribution of 10-30% by region (Fig. S8). Thus, when accounting for the increase in atmospheric moisture in the detection of Antarctic ARs, ..."

LL174-175: change colormap in Fig. 4a,b → especially on the coastal lines, it is difficult to distinguish between dark blue and black

We have tried a number of different colormaps and coastline thicknesses for these figures to maximize the interpretability of the data while using sequential, colorblind-friendly colormaps that are intuitive with respect to the variables plotted (in this case, precipitation), and settled on the blue sequential map as optimal. We recognize that having two sets of 9-panel plots in Figs. 3 and 4, with each subplot containing a full map of Antarctica, means that there may be some loss of regional detail, particularly in a printed version of the article. We hope that with the high resolution versions we provided for each figure, readers can zoom in to look at the details (for example, Fig. 4a):

L183-184: I cannot see the AR-related rainfall in Fig. S7. Do you mean Fig. S9?

Similar to an earlier comment, the reference to Fig. S7 in this sentence refers to "increases in total rainfall", which can be seen in the figure. The idea is that readers can see changes in AR attributed rainfall in Fig. 4d (referenced in previous sentence) and compare to the changes in total rainfall in Fig. S7, if they like. The comparison between Figs. 4b and 4e shows the rise in the relative contribution of ARs to the total rainfall, but comparing S7 and 4d provides a more direct comparison.

L185: The maximum value in Fig. 4e is 24%. Do you mean 21-24%?

These are ice sheet-integrated values, where the relative contribution of AR rainfall to the total rainfall is 21% in the present day. We have revised the text to mention this:

Increases in AR precipitation outpace increases in total precipitation (Fig. S7), such that the relative contribution of ARs to the total rises to 24 +/- 2% (ice sheet-integrated value, see Fig. 3e for regional changes)...

LL259-262: Is this the reason that you use CESM2?

Yes, in part - as mentioned in this section, previous studies such as Zhang et al. (2024, doi: 10.1029/2023JD039359) have shown that there are regional variations in moisture increase over the Southern Ocean, and that the increase is nonlinear and influenced by internal variability in addition to the climate forcing. This acts as clear motivation to use large ensembles from climate models to capture a range of uncertainty in future climatic conditions. We also use the CESM2 large ensemble because it has all of the variables at the temporal resolution needed to assess ARs and their in future climates (synoptic scale), which cannot be captured in monthly output, for example (see response to previous comment for details on variables: "For the future climate, you use the CESM2 LE. Why are you using this model?").

L282: present day → present-day

Thanks, we have corrected it.

L322-323: Why do you know that this resolution is sufficient to resolve ARs? → Is there a reference?

We show in this manuscript that Antarctic AR frequencies in CESM2 in the present-day are comparable to MERRA-2 (Fig. 2). Additionally, we are adding in a reference to Payne et al. (2020, doi: 10.1038/s43017-020-0030-5), which is a review paper on future ARs globally (not including Antarctica) and has a helpful summary of the topic:

"To date, most projection studies have relied on simulations from the Fifth Coupled Model Intercomparison Project (CMIP5), with a horizontal grid spacing of roughly 1.5–2.5° (150–250 km). This resolution is technically high enough to detect ARs but is too coarse to capture the fine-scale features involved within ARs, including interactions with orography."

Our edit to the manuscript is as follows:

This temporal and horizontal grid spacing is sufficient to resolve ARs and associated weather features and precipitation (Payne et al. 2020), ...

Based on results from global AR studies (such as Shields et al. 2023, doi: 10.1029/2022GL102091), we would expect higher resolution models to better represent AR characteristics and precipitation, which is motivation for future studies to implement regional climate models to analyze future Antarctic ARs.

LL323-325: I cannot find a comparison between MERRA2 and ice core records in this paper

Thank you for flagging this, we have fixed the typo in the reference to Medley and Thomas, 2019 (doi: [10.1038/s41558-018-0356-x](https://doi.org/10.1038/s41558-018-0356-x)).

LL349, 351, 354, 372, 376, 385, 408, 431, 436, 445: present day → present-day

Done.

L362: time series for each grid point, or only for significant grid points?

We produce a time series of the relative increase in IWV for each grid point in the domain. As you can see from Fig. 1b, there is a statistically significant increase in IWV everywhere (denoted by the stippling):

LL362-364: Rewrite sentence: Then, for each grid point, we produce a time series of the relative monthly increase in IWV for each ensemble member from 2015 to 2100 compared to the historical climatology

Done, thanks for the recommendation.

L372: I don't understand the calculation of the IWV ratios

Please see response to an earlier comment, where we describe how the IWV ratios (aka smoothed curves of the ensemble 10th percentile, mean, and 90th percentile relative increase in IWV) are calculated:

The method for calculating $IWV_{ratio_{future}}$, and generating the moisture scaling for the AR detection threshold, is described on lines 411-448. Essentially, $IWV_{ratio_{future}}$ represents a scaling curve based on the relative increase in atmospheric moisture at the 10th percentile, ensemble mean, and 90th percentile level. The value of the ratio is determined by the three trendlines in Fig. 1c. The value of the ratio does increase every month, meaning that under the ensemble mean scaling curve, for example, the AR detection threshold in January 2095 will be higher than the threshold in January 2070, in accordance with increasing amounts of atmospheric moisture.

We have revised the text to make it more clear how this ratio is calculated (line 416):

Then, for each grid point, we produce a time series of the relative monthly increase in IWV compared to the historical climatology for each ensemble member from 2015 to 2100. We determine the 10th percentile, mean, and 90th percentile of the relative IWV increase amongst ~~CESM2~~ the ensemble members for each month, to capture the range of variability in future moisture projections in CESM2. To generate smooth IWV scaling curves, we We then take a 10-year running mean of the 10th percentile, ensemble mean, and 90th percentile curves and ~~then~~ calculate a third-degree polynomial trend line for each curve (Fig. 1c). These three curves determine the scaling mechanisms used to modify the AR detection threshold to account for increases in atmospheric moisture in the future climate, and increase over time from 2015-2100 in accordance with atmospheric moisture.

We apply each scaling level - the 10th percentile, ensemble mean, and 90th percentile trend line IWV ratios - to the present day vIVT threshold for AR detection from 2066 to 2100 for each grid point, as follows. Here \$AR_{threshold_{future}}\$ represents the future threshold for ARs, \$IWV_{ratio_{future}}\$ represents the moisture scaling factor (10th percentile, ensemble mean, or 90th percentile relative increase in IWV), and \$vIVT_{threshold_{present}}\$ represents the present-day threshold of vIVT used to detect ARs.

Thus, we develop four thresholds for future Antarctic AR detection ~~using different levels~~ to account for uncertainty in the increase in atmospheric moisture: (1) no scaling (the present day vIVT threshold), (2) the ensemble 10th percentile of the relative increase in IWV among ensemble members (less restrictive than the mean increase), (3) the ensemble mean relative increase in IWV, and (4) the ensemble 90th percentile of the relative IWV increase (more restrictive than the mean increase). We then run the ARDT on CESM2 vIVT from 2066 to 2100 with the updated AR thresholds. For the presentation of results, we primarily focus on the difference between (1) no scaling and (3) scaling by the mean relative increase in IWV, though we provide frequency and precipitation analyses for all scenarios in the supplementary figures. For the frequency analysis

presented in the main text, we compare AR frequencies between the future period (2066-2100) and the present day (1980-2014), and mark regions with statistically significant differences wherever the absolute value of the mean difference in AR frequency between periods exceeds the standard deviation among CESM2 ensemble members.

L398-400: I don't understand this statement. It sounds like you present total precipitation only – but later, you compare between total precipitation and rainfall.

Here we state that we don't present snowfall separately. We have plots showing total precipitation (snowfall + rainfall) and rainfall only, but not snowfall. The reason for this is that the total precipitation variable is useful in producing surface mass balance estimates for the ice sheet (i.e. State of the Climate in 2023, doi: 10.1175/BAMS-D-24-0099.1), as snowfall and rainfall often both contribute to the surface mass balance.

We have revised the text:

We focus on presenting total precipitation and rainfall, and not snowfall individually,...

L401-404: Have you regrid MERRA-2 precip to CESM2 grid? → Yes, you mention it in LL 431-432 → mention it earlier

Starting on line 502, we discuss the method for AR precipitation attribution in CESM2 and MERRA-2. For this step, we do not regrid MERRA-2 to the CESM2 grid. It is only when we assess the precipitation bias of CESM2 relative to MERRA-2, that we regrid the MERRA-2 precipitation output to the CESM2 grid, as explained in the methods.

L450: Have you regrid MERRA-2?

Please see previous response.

L459: between the future period and the present day → difference for CESM2 only?

Yes - see line 524:

Then, we take the difference between the seasonal mean in the future period and the present period for each variable in CESM2.

Caption Fig. 2 LL4-5; Caption Fig. 3 L3; Caption Fig. 4 L2: present day → present-day

Done.

Eq. 3: include T_{sfc} between -2 and 0C

We have updated the equation:

$$\text{rainfall} = \text{total precipitation} * \left(\frac{T_{sfc, -2^{\circ}C \leq T \leq 0^{\circ}C} + 2}{2} \right)$$

Figure 1: It seems that the difference is significant for all grid points. Is this true?

Yes, that is the case.

Figure 2: Caption L2: present → present-day

Done.

Caption L6: scaled by these relative average relative increases in IWV → I don't understand

We have fixed the typo:

(d) Future AR frequency difference in CESM2 (2066 to 2100 minus 1980 to 2014) when the present-day threshold for AR detection is scaled by the ~~the~~ **relative average ensemble mean** relative increase in IWV ~~among all ensemble members~~.

Figure 3 and Figure 4: Caption: After (a), (b), (c),

It is not clear to us what the reviewer's suggested edit is here. We have not made any changes to the manuscript in response to the comment, but would be happy to consider any revisions that the reviewer may want to clarify as part of their next review of our manuscript.

Response to reviewer comments on “Rising atmospheric moisture escalates the future impact of atmospheric rivers in the Antarctic climate system” - second revision

Michelle L. Maclennan, Andrew C. Winters, Christine A. Shields, Rudradutt Thaker, Léonard Barthelemy, Francis Codron, and Jonathan D. Wille

Dear Reviewers,

We would like to thank you for your assessment of our manuscript, and for the constructive feedback and positive comments provided. We have revised the text in accordance with the recommendations from Reviewer #4; otherwise, we have not made any other changes. Overall, your reviews have served to strengthen the manuscript and improve its readability. We thank you for taking the time to provide feedback.

Sincerely,
Michelle Maclennan and co-authors

Reviewer #1

Reviewer Report

I appreciate the lengths the authors have gone to address my concerns, and the edits they have made in relation to my comments, as well as those by other reviewers. In particular, I find the edits to section 2.4 and elsewhere have greatly clarified the nature of the circulation changes and how they relate to AR activity in the Antarctic.

I offer one small clarification regarding one of my previous comments, however I appreciate that the authors tested whether the choice of statistical test was important or not (and as expected, it had no qualitative impact on the nature of the results). My main point was when commenting on the choice of a z- or t-test was that even if all members of the ensemble are used, that does not necessarily represent the true population statistics of the model in question. Again, though, I greatly appreciate the lengths the authors went to within their response to confirm that the results of their analysis were not sensitive to the choice of a z-test or a t-test with 39 degrees of freedom.

As the authors have sufficiently addressed my minor concerns, I find the work to be a worthwhile addition to the journal.

We thank the reviewer for taking the time to review this manuscript a second time, and for providing a clarification on the comment regarding the choice of statistical test in relation to the true population statistics of the model.

Reviewer #2

Reviewer Report

I believe that the authors have correctly addressed most of my comments. I congratulate them on their work. Now I feel comfortable giving my recommendation for publication.

We would like to thank the reviewer for taking the time to review this manuscript a second time, and for providing encouraging feedback.

Reviewer #3

Reviewer Report

I have reviewed the authors' responses to my previous comments and am satisfied with the revised manuscript. This is an important paper and I look forward to seeing it published!

We would like to thank the reviewer for this second review and for the positive comments.

Reviewer #4

Reviewer Report

In this study, the authors assess the sensitivity of Antarctic atmospheric rivers (ARs) to projected changes in atmospheric moisture, focusing on both AR frequency and AR-associated precipitation (and rainfall). For their analysis, they use the MERRA-2 reanalysis product (for the current climate) and the CESM2 LE global climate model (for the current and future climate) under the SSP3.7 scenario. In general, the authors find an increase in both AR frequency and precipitation impacts. However, the results are sensitive to the detection algorithm and the magnitude of the water vapor change. Using today's threshold, AR frequency will increase significantly in the future. In contrast, a higher threshold due to warming and hence higher moisture content shows only a moderate increase. The study concludes by linking shifts in atmospheric circulation to variations in AR activity.

Overall, I found the manuscript well written and the methodology clear. The study provides valuable insights for the AR research community, especially in the context of assessing future changes in Antarctica. However, the results should also be treated with caution because only one climate model was used and, as the authors also write, there are uncertainties in the AR detection algorithm. In this case, I would have liked to see other climate models used. Apart from that, I have three minor comments.

We thank the reviewer for taking the time to review this manuscript a second time and for providing constructive comments. We agree that the next step forward is to perform multi-model comparisons of future Antarctic AR frequencies and associated impacts. We hope that our revision of the text during the previous review to include more information on the biases of CESM2 and the value of using the CESM2 large ensemble, along with the comparison to the E3SMv2 in the supplementary material and the emphasis on uncertainties in future AR activity, provides adequate context for readers on how to interpret our results. Our responses to specific comments are provided below:

L100: -1? Is the minus correct?

We are not quite sure what this comment refers to, but would welcome additional clarification from the reviewer. Line 100 states:

"...than 1 kg m⁻² over the Antarctic continent (Fig. 2e-h), which is characterized by..."

L120 and in the Supplement Figure S3: Is the AR frequency in % or days per year?

The frequency biases discussed in this paragraph (L113-124) refer to the results in Fig. 2a in the main text and Fig. S3 in the supplementary material. Antarctic AR frequency is classically shown in % values (see Wille et al. 2021 Fig. 2), meaning that at each location, ARs occur at a certain percentage of the total timesteps. We have continued this convention in the supplementary material for figures S3 and S5. However, for improved interpretability of Fig. 2a in the main text, we use the units days per year instead of timesteps per year, to show AR frequency in a more relatable form.

L190: up to 30% of the total: Do you mean rainfall? If yes, I would include rainfall – it becomes more clear that you don't mean total precipitation.

Yes, we mean total rainfall, and have revised the text as suggested on line 186:

"... up to 30% of the total rainfall..."